# The Indra Representation Hypothesis for Multimodal Alignment

**Jianglin Lu**[1]* **Hailing Wang**[1]* **Kuo Yang**[1] **Yitian Zhang**[1] **Simon Jenni**[3] **Yun Fu**[1,2]

[1]Department of Electrical and Computer Engineering, Northeastern University
[2]Khoury College of Computer Science, Northeastern University
[3]Adobe Research

## Abstract

Recent studies have uncovered an interesting phenomenon: unimodal foundation models tend to learn convergent representations, regardless of differences in architecture, training objectives, or data modalities. However, these representations are essentially internal abstractions of samples that characterize samples independently, leading to limited expressiveness. In this paper, we propose *The Indra Representation Hypothesis*, inspired by the philosophical metaphor of Indra's Net. We argue that representations from unimodal foundation models are converging to implicitly reflect a shared relational structure underlying reality, akin to the relational ontology of Indra's Net. We formalize this hypothesis using the $\mathcal{V}$-enriched Yoneda embedding from category theory, defining the Indra representation as a relational profile of each sample with respect to others. This formulation is shown to be unique, complete, and structure-preserving under a given cost function. We instantiate the Indra representation using angular distance and evaluate it in cross-model and cross-modal scenarios involving vision, language, and audio. Extensive experiments demonstrate that Indra representations consistently enhance robustness and alignment across architectures and modalities, providing a theoretically grounded and practical framework for training-free alignment of unimodal foundation models. Our code is available at https://github.com/Jianglin954/Indra.

## 1 Introduction

Through large-scale pretraining, foundation models have emerged as a transformative paradigm in artificial general intelligence, demonstrating impressive progress across diverse domains, such as natural language processing [11, 25, 46], computer vision [47, 77, 61, 12], and speech processing [66, 2]. These unimodal models are typically trained on web-scale datasets and acquire generalized representations that can be adapted to a broad range of downstream tasks. The representative models, such as BERT [11] for language, ViT [12] for vision, and Wav2Vec [66] for audio, have demonstrated promising performance within their respective domains.

Since real-world information is inherently multimodal (text, images, and audio frequently co-occur and complement each other), relying on a single modality for understanding is typically insufficient. To extend unimodal foundation models for cross-modal tasks, a growing body of research has exploited strong unimodal encoders as the core components to build multimodal systems [62, 42, 63]. They assume that unimodal models are able to provide specialized, high-quality, and high-level representations for each individual modality, which can then be aligned, fused, or bridged to enable multimodal interactions [41, 45]. A common strategy to achieve multimodal understanding is to align unimodal outputs in a shared representation space through cross-modal objectives [62, 41, 45, 55].

---

*Equal Contribution. Corresponding author: `JianglinLu@outlook.com`.

39th Conference on Neural Information Processing Systems (NeurIPS 2025).

This usually relies on external mechanisms, such as alignment losses, fusion modules, and prompt tuning, thus requiring large-scale datasets and extensive retraining for modality alignment.

Interestingly, recent studies [60, 58, 45, 67] suggest that powerful unimodal models may already exhibit latent cross-modal capabilities, as the representations they produce (when grounded on the same physical entity) tend to describe the same underlying semantics from different sensory perspectives. Previous evidence has revealed that adding only a single linear transformation is capable of bridging an auditory model with a large language model (LLM)[59], integrating a vision model into an LLM [55], or conversely, stitching an LLM towards a vision model [36]. Even without retraining, well-pretrained vision encoders exhibit high semantic similarity with language encoders [52]. Further studies [29, 34, 27, 49] have revealed that models trained on different data modalities converge, as different models are all trying to arrive at a representation of reality. Thus, unimodal models may encode modality-agnostic representations in abstract representational space, even without explicit alignment. While conceptually appealing, the specific form of convergent representations and their eventual convergence targets remain elusive and largely unexplored.

In this paper, we posit that the underlying convergent representation is inherently the Indra Representation—a conceptual abstraction inspired by the philosophical metaphor of Indra's Net. Originating in ancient Buddhist philosophy, Indra's Net describes a vast, infinite web of jewels, each reflecting all others. Every jewel is both a part and a reflection of the whole, suggesting that all phenomena are interdependent, mutually defining, and inherently connected. We draw an analogy between this worldview and the notion of representation convergence, and introduce ***The Indra Representation Hypothesis***: *well-trained unimodal models tend to produce convergent representations that implicitly reflect a shared relational structure underlying reality, echoing the relational ontology of Indra's Net*. In this view, the representation of each entity is not defined in isolation, but rather emerges from its relational context, i.e., its reflections of all other entities.

To explore this hypothesis, we introduce a theoretical definition of the Indra representation grounded in category theory [33]. Specifically, we define it as the $\mathcal{V}$-enriched Yoneda embedding of a sample within a category enriched over a **Cost**-category. This formulation effectively maps each sample to its covariant Hom-functors in the sample category, thereby encoding its relational profile within the structure of the dataset. We provide theoretical guarantees that the Indra representation is unique, complete, and structure-preserving, offering a principled foundation for its effectiveness. In particular, we prove that it uniquely and completely characterizes each sample within the relational structure induced by a given cost function, while preserving essential properties of that structure.

To instantiate this theory in practice, we adopt angular distance as the cost function, yielding a simple yet powerful realization of the Indra representation. This concrete formulation enables empirical evaluation and allows us to investigate how Indra representations can uncover and support the latent cross-modal capabilities of unimodal models. We validate our approach across a range of scenarios involving cross-model and cross-modal understanding, including single-modality settings, vision-language pairs, and speech-language pairs. Extensive experiments demonstrate the effectiveness and generality of the proposed Indra representation across different architectures and modalities.

## 2 Preliminaries

### 2.1 Indra's Net

Indra's Net is a philosophical metaphor originating in ancient Indian and Mahāyāna Buddhist thought, particularly from the Avataṃsaka Sūtra [9, 22]. It is used to symbolize the universe as a web of interdependent connections among all of its members, expressing the concept of interconnectedness, non-duality, and the interpenetration of all phenomena. Francis H. Cook describes it as [10]:

> *Imagine a vast, infinite cosmic net belonging to the god Indra. At each node of the net is a jewel or crystal. Each jewel reflects every other jewel in the net, and in each of those reflections are further reflections of all other jewels, recursively and infinitum.*

The metaphor of Indra's Net resonates deeply with the foundational principles across diverse disciplines. For instance, Gergen et al. [17] posit that identities, thoughts, and actions are not solely products of isolated minds but are co-constructed through interactions and relationships with others. Markus et al. [53] propose the theory of interdependent self-construal, emphasizing that the self is

defined relationally through one's social roles, group memberships, and interpersonal obligations. In physics, field theory [14, 54] illustrates a relational structure where the field at any point depends on all sources throughout space, a concept reminiscent of Indra's Net, in which each jewel reflects and is reflected by all others. Similarly, modern particle physics [18, 4] reveals the properties of an elementary particle through its interactions with other particles. In linguistics, the linguistic principle articulated by J.R. Firth [15], "You shall know a word by the company it keeps", asserts that a word's meaning is derived from its co-occurrence with other words. This principle forms the basis for modern language models like Word2Vec [56] and its successors. Furthermore, Kasulis [32] suggests DNA as a better image of Indra's Net, where every cell contains the blueprint for the whole organism.

## 2.2 Representation Convergence

Recent studies [29, 76, 20, 74, 40] have revealed a striking phenomenon: unimodal foundation models tend to learn convergent representations, regardless of their architectures, training objectives, or data modalities. For example, Bürger et al. [5] show that a two-dimensional representation of truth emerges universally across LLMs of varying sizes and from different model families. Roeder et al. [65] prove that a wide class of discriminative and autoregressive models are identifiable in function space up to a linear transformation. Tan et al. [68] find strong correlations in both in-distribution and out-of-distribution steerability between LLaMA [69] and Qwen [3]. Huh et al. [29] attribute this convergence to a shared goal: approximating an underlying representation of reality. Khosla et al. [34] argue that both artificial and biological systems converge toward representations that capture the causal structure of the world. Hosseini et al. [27] further observe that high-performing artificial neural networks and biological brains tend to develop similar internal representations under naturalistic training conditions. Additional evidence and analyses on representation convergence can be found in the comprehensive survey [49].

Despite this emerging consensus, it remains unclear how these representations converge and what they ultimately converge to. Notably, prior studies rely primarily on model outputs (embeddings) as proxies for representations, but these representations suffer from ① structural myopia: representations are typically treated as isolated carriers of information, ignoring structural interrelations within the broader data manifold; ② limited expressiveness: unimodal representations often exhibit inferior quality in matching and alignment compared to those from multimodal foundation models; and ③ dimensional incompatibility: representations across models and modalities often differ in dimensionality, thus requiring additional post-processing for cross-modal matching. In light of these challenges, *we argue that representations from model outputs do not reflect the final converged form, but instead serve as the foundation upon which such a form can be built.* In the next section, we introduce a novel representation hypothesis inspired by the metaphor of Indra's Net to hypothesize the concrete structure of convergent representations and illuminate what they ultimately converge to.

# 3 Methodology

## 3.1 The Indra Representation Hypothesis

In this paper, we advocate a shift in perspective on representation convergence, inspired by the metaphor of Indra's Net: a sample should be represented not in isolation, but through its pattern of relationships to other samples. In this view, representations emerge from a structure of mutual interdependence. We formalize this perspective through the Indra Representations Hypothesis:

> **The Indra Representation Hypothesis**: *Neural networks, trained with different objectives on different data and modalities, tend to learn convergent representations that implicitly reflect a shared relational structure underlying reality—parallel to the relational ontology of Indra's Net.*

This hypothesis posits that unimodal foundation models, after extensive pretraining, tend to produce representations that converge to capture the inherent relational structure of reality—a structure characterized by interdependence, contextuality, and mutual influence. However, current methods that treat model outputs as final representations fail to reflect this structure. These methods typically emphasize individual embedding information while neglecting the crucial relational patterns between samples. In the next section, we introduce the *Indra Representation*, a novel representation framework inspired by the metaphor of Indra's Net, to reveal the underlying relational structure. In this framework,

representations are not independent embeddings but mutually reflective entities woven into a web of interdependent relationships, revealing the deeper relational structure that underpins the data.

## 3.2 From Metaphor to Theory

Indra's Net is a philosophical metaphor that symbolizes the interconnectedness of the universe. It aligns with the foundational principles in modern science, as mentioned in Section 2.1. To translate this philosophical insight into representation learning, we first introduce the Yoneda Lemma and its corollary, which provide the theoretical foundation for defining our proposed Indra representation.

**Lemma 1** (Yoneda Lemma [33, 64]). *Let $\mathcal{C}$ be a locally small category, $A$ be an object in $\mathcal{C}$, and $F : \mathcal{C} \to \textbf{Set}$ be a functor from $\mathcal{C}$ to the category of sets. Then, there exists a bijection, natural in both $A$ and $F$, between the set of natural transformations from the hom-functor $h_A = Hom_{\mathcal{C}}(A, -)$ to $F$, and the set $F(A)$. This bijection is given by:*

$$Nat(h_A, F) \cong F(A). \tag{1}$$

**Corollary 1** (Yoneda Embedding [33, 64]). *For any two objects $A, B$ in a locally small category $\mathcal{C}$, there is a bijection:*

$$Nat(Hom_{\mathcal{C}}(A, -), Hom_{\mathcal{C}}(B, -)) \cong Hom_{\mathcal{C}}(B, A). \tag{2}$$

*This demonstrates that the functor $Y : \mathcal{C}^{op} \to [\mathcal{C}, \textbf{Set}]$, defined by $Y(A) = h_A = Hom_{\mathcal{C}}(A, -)$, is fully faithful. This functor $Y$ is known as the Yoneda embedding.*

The Yoneda Lemma provides a profound understanding of how an object in a category is characterized by its relationships (morphisms) with all other objects, rather than by its internal properties. Its corollary further shows that any locally small category $\mathcal{C}$ can be embedded into a category of presheaves on $\mathcal{C}$. To introduce our Indra representations, we give the following definitions:

**Definition 1** (Sample Category). *Let $\mathcal{X}$ be a set of samples, possibly infinite. The sample category $\mathcal{C}$ enriched over the $\textbf{Cost}$-category $\mathcal{V} = ([0, \infty], \geq, 0, +)$ consists of: ① Objects: $Ob(\mathcal{C}) = \mathcal{X}$. ② Hom-objects: for every $X_i, X_j \in Ob(\mathcal{C})$, the hom-object $\mathcal{C}(X_i, X_j)$ is given by a cost function $d(X_i, X_j) \in [0, \infty]$, which is an object in $\mathcal{V}$. ③ Identity: for all $X_i \in Ob(\mathcal{C})$, the identity morphism $id_{X_i} : I \to \mathcal{C}(X_i, X_i)$ in $\mathcal{V}$ is $0 \to d(X_i, X_i)$, where $d(X_i, X_i) = 0$. ④ Composition: for all $X_i, X_j, X_k \in Ob(\mathcal{C})$, the composition morphism $M_{X_i, X_j, X_k} : \mathcal{C}(X_j, X_k) \otimes \mathcal{C}(X_i, X_j) \to \mathcal{C}(X_i, X_k)$ in $\mathcal{V}$ is: $d(X_j, X_k) + d(X_i, X_j) \to d(X_i, X_k)$. This morphism exists in $\mathcal{V}$ if and only if $d(X_j, X_k) + d(X_i, X_j) \geq d(X_i, X_k)$, which is precisely the triangle inequality.*

**Definition 2** ($\mathcal{V}$-enriched Yoneda embedding). *Let $[\mathcal{C}^{op}, \mathcal{V}]$ be the category of $\mathcal{V}$-presheaves on $\mathcal{C}$. The $\mathcal{V}$-enriched Yoneda embedding is a $\mathcal{V}$-functor $Y : \mathcal{C} \to [\mathcal{C}^{op}, \mathcal{V}]$. For each object $X_i \in Ob(\mathcal{C})$, $Y(X_i)$ is the $\mathcal{V}$-presheaf $h_{X_i} : \mathcal{C}^{op} \to \mathcal{V}$ defined by: $h_{X_i}(X_j) = \mathcal{C}(X_j, X_i) = d(X_j, X_i)$ for any $X_j \in \mathcal{C}^{op}$. For every $X_i, X_j \in Ob(\mathcal{C})$, $Y$ defines a map $Y_{X_i, X_j} : \mathcal{C}(X_i, X_j) \to [\mathcal{C}^{op}, \mathcal{V}](Y(X_i), Y(X_j))$.*

**Theorem 1.** *The $\mathcal{V}$-enriched Yoneda embedding $Y : \mathcal{C} \to [\mathcal{C}^{op}, \mathcal{V}]$ for the sample category $\mathcal{C}$ enriched over $\mathcal{V} = ([0, \infty], \geq, 0, +)$ with the cost function $d$ is $\mathcal{V}$-fully faithful.*

The sample category $\mathcal{C}$ actually forms a Lawvere metric space [39], and the corresponding $\mathcal{V}$-enriched Yoneda embedding maps each sample $X_i$ to a functor $Y(X_i)$, which captures the cost profile from all other samples to $X_i$. Theorem 1 shows that the sample category can be fully and faithfully represented within a category of $\mathcal{V}$-presheaf, preserving its entire structure including its metric information. In other words, each sample $X_i$ can be uniquely represented by its cost vector $d(\cdot, X_i)$ with all samples. Based on these, we introduce the Indra representation and state the following theorems:

**Definition 3** (**Indra Representation**). *For each sample $X_i \in Ob(\mathcal{C})$, we define its Indra representation as the $\mathcal{V}$-functor $\mathcal{C}(X_i, -)$, i.e., the collection of values obtained by evaluating it under the (covariant) $\mathcal{V}$-enriched Yoneda embedding on all objects of the category $\mathcal{C}$.*

**Proposition 1.** *If two samples $X_i, X_j \in Ob(\mathcal{C})$ have $\mathcal{V}$-naturally isomorphic Indra representations and the cost function $d$ satisfies the $T_0$ separation axiom, then $X_i = X_j$.*

**Theorem 2.** *For any $\mathcal{V}$-functor $P : \mathcal{C} \to \mathcal{V}$, the $\mathcal{V}$-hom-object of $\mathcal{V}$-natural transformations from the Indra representation of sample $X_i$ to $P$, denoted by $[\mathcal{C}, \mathcal{V}](\mathcal{C}(X_i, -), P)$, is $\mathcal{V}$-isomorphic to $P(X_i)$.*

**Corollary 2.** *The relational structure among objects in the sample category $\mathcal{C}$ is preserved and reflected in the relationships between their Indra representations.*

The Indra representation of a sample $X_i$, defined as the $\mathcal{V}$-functor $\mathcal{C}(X_i, -)$, can be interpreted as a relational profile of $X_i$, that is, the cost from $X_i$ to every other sample in the category. Proposition 1 establishes that the Indra representation is a faithful encoding: no two distinct samples share the same representation. Theorem 2 further shows that the Indra representation is complete, in the sense that it encapsulates all the information needed to determine how distances from $X_i$ behave under any admissible distance assignment. Furthermore, Corollary 2 demonstrates that the relationships between samples are in one-to-one correspondence with the relationships between their Indra representations.

### 3.3 Instantiation of Indra Representation

We now demonstrate how to instantiate the Indra representation for a real dataset $\mathcal{X} = \{X_1, \ldots, X_n\}$ consisting of $n$ samples. We define the object set of the enriched category as $\mathrm{Ob}(\mathcal{C}) = \mathcal{X}$ and specify the hom-object $\mathcal{C}(X_i, X_j)$ as the cost $d(X_i, X_j)$ between samples $X_i$ and $X_j$, $\forall X_i, X_j \in \mathrm{Ob}(\mathcal{C})$. To define a valid Indra representation, the cost function $d$ must satisfy two properties: ① $d(X_i, X_i) = 0$ for $\forall X_i \in \mathcal{X}$; and ② $d(X_i, X_k) \leq d(X_i, X_j) + d(X_j, X_k)$ for $\forall X_i, X_j, X_k \in \mathcal{X}$. Several distance metrics satisfy these conditions. In this work, we adopt a simple yet effective choice by defining $d(X_i, X_j)$ as the angular distance between the embeddings of $X_i$ and $X_j$:

$$d(X_i, X_j) := \arccos\left(\frac{f(X_i)^\top f(X_j)}{\|f(X_i)\|\|f(X_j)\|}\right), \quad \forall X_i, X_j \in \mathcal{X} \tag{3}$$

where $f : \mathcal{X} \to \mathbb{R}^{d^{\mathcal{X}}}$ is a modality-specific foundation model, $f(X_i)$ denotes the internal representation of $X_i$ produced by $f$, and $d^{\mathcal{X}}$ is the output dimensionality of the model. The use of angular distance ensures that $d$ defines a valid Lawvere cost function. Given this cost function, the Indra representation of each sample $X_i \in \mathcal{X}$ is the covariant Hom-functor:

$$\mathcal{C}(X_i, -) : \mathcal{X} \to [0, \infty], \quad X_j \mapsto d(X_i, X_j). \tag{4}$$

This distance-based embedding forms a principled and interpretable representation. In the finite case, this can be written as $\mathcal{C}(X_i, -) = [d(X_i, X_1), \ldots, d(X_i, X_n)]$, which captures the relational profile of $X_i$ with respect to all other samples. Notably, when the relational profile of each sample is computed using only a small set of representative landmarks, this formulation closely aligns with the approach proposed in [57].

### 3.4 Relational Matching across Modalities

Our hypothesis in Section 3.1 posits that unimodal foundation models learn convergent representations that capture the shared relational structure underlying reality. The proposed Indra representations are designed to reflect this inherent structure and can thus be leveraged to improve cross-modal understanding. To demonstrate how Indra representations facilitate relational matching across modalities, we consider a dataset $\mathcal{D} = \{(U_i, Q_i)\}_{i=1}^n$ of $n$ samples, where $U_i \in \mathcal{U}$ and $Q_i \in \mathcal{Q}$ correspond to instances from two distinct modalities, and $\mathcal{D} \subseteq \mathcal{U} \times \mathcal{Q}$. For single-modality scenarios, we define $U_i = Q_i, \forall i \in \{1, \ldots, n\}$. We use two pretrained foundation models $f : \mathcal{U} \to \mathbb{R}^{d^{\mathcal{X}}}$ and $g : \mathcal{Q} \to \mathbb{R}^{d^{\mathcal{Y}}}$ to extract modality-specific embeddings, where $d^{\mathcal{X}}$ and $d^{\mathcal{Y}}$ are the embedding dimensionalities of the two models, respectively. Based on these embeddings, we construct the Indra representations $\mathbf{I}^{\mathcal{U}}$ and $\mathbf{I}^{\mathcal{Q}}$ for each modality as follows:

$$\mathbf{I}^{\mathcal{U}}_{ij} = d(U_i, U_j), \quad \mathbf{I}^{\mathcal{Q}}_{ij} = d(Q_i, Q_j), \quad \forall i, j \in \{1, \ldots, n\}, \tag{5}$$

where the cost function $d$ is defined in Equation 3. In practice, we may apply post-processing operations such as sparsification and normalization to enhance robustness:

$$\hat{\mathbf{I}}^{\mathcal{U}} = \mathtt{operator}(\mathbf{I}^{\mathcal{U}}), \quad \hat{\mathbf{I}}^{\mathcal{Q}} = \mathtt{operator}(\mathbf{I}^{\mathcal{Q}}), \tag{6}$$

where $\mathtt{operator}(\cdot)$ denotes the chosen post-processing function. Unlike traditional representation approaches that reflect only internal characteristics of samples, the proposed Indra representations act as external representations, where each vector captures interdependencies by encoding the sample's relative profile within the dataset.

## 4 Experiments

To comprehensively assess the effectiveness of our Indra representation, we perform evaluations across a range of settings, including unimodal vision, vision–language, and speech–language tasks.

Table 1: Accuracy (%) on CIFAR-10 and CIFAR-100 under different Gaussian noise levels.

| CIFAR-10 | $\sigma$=0.0 | $\sigma$=3.0 | $\sigma$=5.0 | $\sigma$=7.0 | CIFAR-100 | $\sigma$=0.0 | $\sigma$=3.0 | $\sigma$=5.0 | $\sigma$=7.0 |
|---|---|---|---|---|---|---|---|---|---|
| ViT | 93.98 | 87.75 | 79.77 | 68.15 | ViT | 79.45 | 54.69 | 35.76 | 27.45 |
| Indra | 94.84 | 89.51 | 80.84 | 68.71 | Indra | 80.09 | 69.00 | 51.59 | 32.74 |
| Convnext | 97.00 | 85.89 | 80.10 | 65.85 | Convnext | 85.77 | 62.79 | 34.39 | 21.28 |
| Indra | 97.21 | 92.86 | 81.59 | 66.64 | Indra | 85.64 | 72.16 | 51.51 | 30.25 |
| Dinov2 | 99.19 | 95.21 | 85.57 | 76.54 | Dinov2 | 91.97 | 82.21 | 63.06 | 40.16 |
| Indra | 99.14 | 96.87 | 89.73 | 77.92 | Indra | 91.93 | 84.83 | 74.29 | 58.67 |

Table 2: Accuracy (%) on Office-Home dataset under different Gaussian noise levels.

| Art | $\sigma$=0.0 | $\sigma$=3.0 | $\sigma$=5.0 | $\sigma$=7.0 | Clipart | $\sigma$=0.0 | $\sigma$=3.0 | $\sigma$=5.0 | $\sigma$=7.0 |
|---|---|---|---|---|---|---|---|---|---|
| ViT | 80.25 | 64.40 | 44.03 | 22.63 | ViT | 73.20 | 50.40 | 28.64 | 15.23 |
| Indra | 79.63 | 65.02 | 43.62 | 27.57 | Indra | 69.76 | 54.98 | 33.10 | 18.21 |
| Convnext | 89.71 | 62.76 | 27.98 | 12.14 | Convnext | 83.62 | 54.07 | 20.85 | 09.74 |
| Indra | 87.86 | 59.88 | 28.81 | 14.20 | Indra | 82.70 | 57.85 | 25.09 | 11.34 |
| Dinov2 | 87.65 | 73.05 | 46.91 | 27.78 | Dinov2 | 88.43 | 75.14 | 51.09 | 31.04 |
| Indra | 87.04 | 70.99 | 47.53 | 27.37 | Indra | 87.29 | 76.63 | 54.75 | 33.56 |
| **Product** | $\sigma$=0.0 | $\sigma$=3.0 | $\sigma$=5.0 | $\sigma$=7.0 | **Real** | $\sigma$=0.0 | $\sigma$=3.0 | $\sigma$=5.0 | $\sigma$=7.0 |
| ViT | 92.34 | 80.74 | 61.15 | 35.25 | ViT | 89.22 | 82.11 | 60.09 | 35.32 |
| Indra | 89.75 | 81.53 | 64.08 | 40.77 | Indra | 87.16 | 83.49 | 63.65 | 40.48 |
| Convnext | 96.62 | 84.91 | 44.26 | 19.37 | Convnext | 93.46 | 82.11 | 38.30 | 17.78 |
| Indra | 96.73 | 85.92 | 45.61 | 22.18 | Indra | 93.35 | 84.63 | 40.71 | 19.61 |
| Dinov2 | 96.73 | 93.24 | 83.33 | 60.70 | Dinov2 | 92.78 | 87.39 | 71.44 | 48.51 |
| Indra | 96.40 | 92.79 | 84.46 | 60.59 | Indra | 92.89 | 88.53 | 73.17 | 49.89 |

## 4.1 Evaluation on Single Modality

*Datasets.* We first conduct classification tasks on the CIFAR-10 [37], CIFAR-100 [37], and Office-Home [72] datasets. For CIFAR-10 and CIFAR-100, we use the standard data splits. For Office-Home, we evaluate classification accuracy across four distinct domains: Art, Clipart, Product, and Real-World, using an $80/20$ split for training and testing. Each domain exhibits unique visual styles and distribution shifts, making the dataset a widely used benchmark for evaluating the robustness and generalization of vision models in object recognition tasks. Across all datasets, we adopt logistic regression (i.e., linear probing) to assess the quality of the extracted representations.

*Foundation Models.* For vision models, we evaluate ViT [12], Convnext [77], and Dinov2 [61].

*Evaluation Metrics.* We assess model classification accuracy (%) using ground-truth labels. To investigate the robustness of Indra representations, we inject Gaussian noise into the features with varying standard deviations $\sigma \in \{0.0, 3.0, 5.0, 7.0\}$. For each noise level, we perturb the features accordingly and train a linear classifier on the noisy representations. This allows us to assess how classification performance degrades as the feature representations are increasingly corrupted by noise.

*Analysis.* In Tables 1 and 2, we report the classification results on the CIFAR-10, CIFAR-100, and Office-Home datasets. The results clearly show that stronger backbone models (e.g., Dinov2) lead to better performance for Indra representations across all noise levels. For instance, on CIFAR-100 with $\sigma$=0.0, our Indra representations achieve 91.93% accuracy using Dinov2 as the backbone, compared to 85.64% with Convnext and 80.09% with ViT. This performance gap persists and even widens under higher noise: at $\sigma = 7.0$, Indra representations with Dinov2 maintain 58.67%, while Convnext and ViT drop to 30.25% and 32.74%, respectively. In addition, as Gaussian noise

Table 3: Performance on image-text datasets $\mathcal{D}$ using different representations $\mathcal{R}$ (`Orign`: original, `Indra`: Indra representation) with various vision (`Vis-E`) and language (`Lan-E`) models.

| $\mathcal{D}$ | Vis-E | Lan-E | $\mathcal{R}$ | Top-5 | | Top-10 | | Top-30 | | Top-50 | |
|---|---|---|---|---|---|---|---|---|---|---|---|
| | | | | T→I | I→T | T→I | I→T | T→I | I→T | T→I | I→T |
| MS-COCO | CLIP-I | CLIP-T | Orign | 1.420 | 1.381 | 2.734 | 2.661 | 7.634 | 7.470 | 12.212 | 11.986 |
| | ViT | BERT | Orign | 0.482 | 0.483 | 0.967 | 0.966 | 2.911 | 2.905 | 4.863 | 4.846 |
| | | | Indra | 0.663 | 0.832 | 1.303 | 1.613 | 3.787 | 4.426 | 6.199 | 7.036 |
| | ViT | Roberta | Orign | 0.486 | 0.491 | 0.970 | 0.981 | 2.912 | 2.927 | 4.853 | 4.874 |
| | | | Indra | 1.048 | 0.880 | 2.065 | 1.749 | 5.970 | 5.149 | 9.702 | 8.446 |
| | Convnext | BERT | Orign | 0.396 | 0.474 | 0.837 | 0.950 | 2.603 | 2.851 | 4.412 | 4.755 |
| | | | Indra | 0.612 | 0.537 | 1.127 | 1.022 | 3.182 | 2.875 | 5.242 | 4.783 |
| | Convnext | Roberta | Orign | 0.492 | 0.480 | 0.985 | 0.964 | 2.962 | 2.889 | 4.940 | 4.824 |
| | | | Indra | 1.005 | 0.616 | 1.930 | 1.217 | 5.247 | 3.538 | 8.267 | 5.790 |
| | Dinov2 | BERT | Orign | 0.496 | 0.473 | 0.991 | 0.947 | 2.969 | 2.852 | 4.936 | 4.760 |
| | | | Indra | 0.540 | 0.539 | 1.123 | 1.022 | 3.194 | 2.872 | 5.277 | 4.804 |
| | Dinov2 | Roberta | Orign | 0.468 | 0.490 | 0.945 | 0.982 | 2.859 | 2.949 | 4.779 | 4.914 |
| | | | Indra | 1.016 | 0.949 | 1.978 | 1.863 | 5.603 | 5.370 | 9.021 | 8.766 |
| NOCAPS | CLIP-I | CLIP-T | Orign | 1.357 | 1.325 | 2.556 | 2.499 | 6.860 | 6.717 | 10.795 | 10.584 |
| | ViT | BERT | Orign | 0.479 | 0.474 | 0.956 | 0.947 | 2.864 | 2.844 | 4.769 | 4.742 |
| | | | Indra | 0.701 | 0.667 | 1.375 | 1.293 | 3.960 | 3.712 | 6.449 | 6.069 |
| | ViT | Roberta | Orign | 0.484 | 0.483 | 0.966 | 0.963 | 2.891 | 2.886 | 4.814 | 4.805 |
| | | | Indra | 0.924 | 0.727 | 1.792 | 1.419 | 5.014 | 4.102 | 8.011 | 6.700 |
| | Convnext | BERT | Orign | 0.449 | 0.451 | 0.904 | 0.910 | 2.754 | 2.752 | 4.640 | 4.604 |
| | | | Indra | 0.415 | 0.485 | 0.906 | 0.971 | 2.911 | 2.907 | 4.821 | 4.845 |
| | Convnext | Roberta | Orign | 0.472 | 0.462 | 0.944 | 0.926 | 2.833 | 2.781 | 4.721 | 4.647 |
| | | | Indra | 0.764 | 0.557 | 1.472 | 1.102 | 4.079 | 3.338 | 6.494 | 5.526 |
| | Dinov2 | BERT | Orign | 0.465 | 0.439 | 0.928 | 0.883 | 2.701 | 2.674 | 4.500 | 4.485 |
| | | | Indra | 0.566 | 0.485 | 1.065 | 0.971 | 3.179 | 2.907 | 5.238 | 4.845 |
| | Dinov2 | Roberta | Orign | 0.497 | 0.467 | 0.993 | 0.936 | 2.969 | 2.822 | 4.938 | 4.712 |
| | | | Indra | 0.830 | 0.774 | 1.604 | 1.513 | 4.549 | 4.324 | 7.335 | 7.019 |

increases, our Indra representations consistently retain higher classification accuracy compared to the original representations, highlighting their robustness in the classification tasks. The performance gains of Indra representations hold across multiple backbone architectures (i.e., `ViT`, `Convnext`, and `Dinov2`), indicating the broad applicability of the proposed method.

## 4.2 Evaluation on Vision & Language Modalities

*Datasets.* We adopt two widely used image-text datasets: MS-COCO [44] and NOCAPS [1] to evaluate performance on vision and language modalities. MS-COCO serves as a standard benchmark for image captioning and retrieval tasks, while NOCAPS poses a greater challenge due to its focus on novel object categories. We use the validation sets of both datasets for evaluation.

*Foundation Models.* We use the same vision models as in Section 4.1. For language models, we evaluate `BERT` [11] and `Roberta` [46], both of which are pretrained independently on unimodal data without cross-modal alignment. We include CLIP [62] as the aligned baseline for evaluation.

*Evaluation Metrics*: We adopt CLIPScore [26] as the evaluation metric, which measures semantic alignment between image and text based on the cosine similarity of their embeddings within the CLIP multimodal space. We report Top-$k$ matching accuracy ($k \in \{5, 10, 30, 50\}$) in both text-to-image (T→I) and image-to-text (I→T) tasks.

*Analysis.* Table 3 compares the performance of original embeddings versus Indra representations across both datasets using various combinations of vision and language models. The results clearly demonstrate that Indra representations lead to consistent performance gains across different archi-

Table 4: Performance on audio-text dataset $\mathcal{D}$ using different representations $\mathcal{R}$ (`Orign`: original, `Indra`: Indra representation) with various audio (`Aud-E`) and language (`Lan-E`) models, where `*-b` and `*-l` refer to the base and large versions of model `*`, respectively.

| $\mathcal{D}$ | Aud-E | Lan-E | $\mathcal{R}$ | Top-5 | | Top-10 | | Top-30 | | Top-50 | |
|---|---|---|---|---|---|---|---|---|---|---|---|
| | | | | T→A | A→T | T→A | A→T | T→A | A→T | T→A | A→T |
| TIMIT | CLAP-I | CLAP-T | Orign | 1.062 | 1.836 | 2.046 | 3.611 | 5.726 | 10.146 | 9.204 | 16.225 |
| | wav2vec-b | Roberta | Orign | 0.319 | 0.072 | 0.670 | 0.276 | 2.214 | 1.409 | 3.655 | 2.750 |
| | | | Indra | 0.413 | 0.578 | 0.819 | 1.209 | 2.506 | 2.953 | 4.225 | 4.692 |
| | wav2vec-l | Roberta | Orign | 0.418 | 0.308 | 0.864 | 0.634 | 2.548 | 2.194 | 4.200 | 3.738 |
| | | | Indra | 0.363 | 0.578 | 0.908 | 1.243 | 2.783 | 2.956 | 4.640 | 4.318 |
| | wavlm-b | Roberta | Orign | 0.328 | 0.328 | 0.659 | 0.648 | 2.080 | 1.920 | 3.518 | 3.208 |
| | | | Indra | 0.436 | 0.578 | 0.913 | 1.241 | 2.493 | 2.976 | 4.227 | 4.338 |
| | wavlm-l | Roberta | Orign | 0.472 | 0.426 | 0.912 | 0.876 | 2.450 | 2.518 | 3.997 | 4.148 |
| | | | Indra | 0.504 | 0.578 | 0.971 | 1.229 | 2.693 | 2.967 | 4.387 | 4.335 |
| | hubert-b | Roberta | Orign | 0.280 | 0.431 | 0.553 | 0.793 | 1.978 | 2.069 | 3.282 | 3.240 |
| | | | Indra | 0.322 | 0.578 | 0.697 | 1.232 | 2.230 | 2.971 | 3.880 | 4.369 |
| | hubert-l | Roberta | Orign | 0.449 | 0.308 | 0.861 | 0.638 | 2.475 | 1.949 | 4.119 | 3.286 |
| | | | Indra | 0.454 | 0.578 | 0.878 | 1.248 | 2.610 | 3.003 | 4.461 | 4.255 |

tectures and modalities. Significant improvements are observed in both T→I and I→T retrieval, highlighting the effectiveness of our method for cross-modal alignment. These findings suggest that the Indra representation offers a generalizable mechanism to improve vision-language matching, independent of model architecture or dataset. Nonetheless, there remains a noticeable gap in performance compared to the fully aligned CLIP model, indicating further room for improvement.

### 4.3 Evaluation on Speech & Language Modalities

*Datasets.* We adopt the TIMIT dataset [16] for audio and language modality experiments. TIMIT contains recordings from 630 speakers representing eight major dialect regions of American English, each reading ten phonetically rich sentences. The dataset provides time-aligned phonetic and word-level transcriptions along with $16k$Hz audio recordings.

*Foundation Models.* For audio models, we evaluate `wav2vec` [66], `wavlm` [6], and `hubert`[28], using both base and large variants. For the language modality, we use `Roberta` [46]. All models are pretrained independently on unimodal data, without any cross-modal alignment. As an aligned baseline, we include CLAP [78], an audio-language model jointly trained on paired audio-text data.

*Evaluation Metrics*: Similarly, we adopt CLAPScore [78] as the evaluation metric. We report Top-$k$ matching ($k \in \{5, 10, 30, 50\}$) in both text-to-audio (T→A) and audio-to-text (A→T) tasks.

*Analysis.* Table 4 presents the audio-text matching results using different audio models. As shown, the Indra representations consistently improve matching performance in both directions across all model configurations. However, compared to the vision-language setting, the improvements in the audio-language modality are relatively modest. This is likely due to the comparatively weaker capacity of the audio models used. Nonetheless, we observe that larger audio models yield better matching accuracy, further supporting the notion that model capacity positively influences cross-modal alignment.

## 5 Related Work

**Instance-Level Representation Learning.** In conventional deep learning paradigms, each data instance is independently encoded into a fixed-dimensional vector, optimized using either supervised signals or unsupervised objectives. In supervised learning, representations are shaped by categorical labels [38, 24], whereas in unsupervised settings, objectives such as reconstruction (e.g., autoencoders [73]) or self-prediction (e.g., BERT [11]) guide the learning of instance-level representations. These approaches primarily focus on learning standalone embeddings and do not explicitly model the relationships between samples, either in training or inference stages.

**Contrastive Representation Learning.** Contrastive methods such as SimCLR [7], MoCo [23], and BYOL [19] introduce pairwise relational inductive bias during training by encouraging the embeddings of similar (augmented) views to be close, while pushing apart dissimilar ones. Building on this framework, CLIP [62] extends contrastive learning to vision-language pretraining by aligning paired image-text embeddings within a shared multimodal space, enabling strong zero-shot performance. BLIP [42] further integrates contrastive and generative objectives to enhance cross-modal representation learning, achieving state-of-the-art results on several vision-language benchmarks. Despite these advances, the sample representations in contrastive frameworks remain inherently instance-centric during inference. Additionally, contrastive learning approaches typically require large-scale datasets to achieve satisfactory performance, as the training objectives rely heavily on diverse and abundant positive and negative pairs.

**Graph-based Representation Learning.** Graph neural networks [35] offer a framework for encoding relational information through message passing across graph-structured data. Variants such as GraphSAGE [21] and GAT [71] have improved flexibility and representation ability. However, most graph-based approaches rely on predefined adjacency structures or proximity assumptions [51], which may embed inductive biases that are misaligned with the latent semantics of the data. They typically operate over local $k$-hop neighborhoods [48], limiting their ability to capture long-range dependencies unless deeply stacked, which can lead to oversmoothing and degraded performance [43].

**Attention-based Representation Learning.** Transformer-based architectures [70] offer a powerful alternative by leveraging global self-attention to aggregate contextual information. Numerous studies [11, 46, 12, 25, 61, 63] have demonstrated that attention-based architectures can be highly effective even in non-sequential domains. While attention enables global interactions both during training and inference, the resulting token-level representations are determined through dynamic mixing rather than explicitly encoding pairwise or global sample-to-sample relationships, making their geometric interpretation less transparent.

In contrast to the above paradigms, our approach explicitly constructs each representation as a relational profile, specifically, a distance-based reflection of its relationship to all other samples. This design is motivated by the philosophical ontology of Indra's Net, where each entity reflects and is reflected by all others, forming a holistic network of interdependence. More importantly, our Indra representation is formally grounded in the Yoneda Lemma from enriched category theory, offering a theoretically sound and interpretable framework for relational representation learning.

# 6   Conclusion

In this paper, we present a theoretical and empirical investigation into the convergent behavior of unimodal foundation models. Motivated by the philosophical metaphor of Indra's Net and grounded in enriched category theory, we introduce the Indra representation as a relational encoding that reflects each sample through its relationships with all others. We demonstrate that this representation can be derived via the $\mathcal{V}$-enriched Yoneda embedding and instantiated practically using angular distance. Our theoretical analysis proves that the proposed Indra representations are unique, complete, and structure-preserving, offering a principled basis for training-free alignment of various foundation models. Through extensive experiments across single-modality, vision-language, and speech-language settings, we demonstrate that Indra representations improve robustness and latent cross-modal capabilities of unimodal foundation models, offering a general framework for training-free multimodal alignment. Our findings suggest a new perspective for bridging modalities, which emphasizes the importance of the intrinsic relational structure of data or reality.

*Limitations.* Constructing exact Indra representations requires a computational complexity of $\mathcal{O}(n^2 d)$ and a memory complexity of $\mathcal{O}(n^2)$ for a dataset with $n$ samples and embedding dimension $d$. This quadratic scaling potentially limits the direct applicability of the exact Indra representations to large-scale settings. However, the scalability concern is addressable in practice. In the literature, there exists a rich body of work on approximating pairwise distances efficiently. For example, approximate nearest neighbor search [30, 31], landmark-based approximation [50, 57], hashing-based methods [8], and sparse graph constructions [75]. From an application perspective, these techniques can be readily adapted to approximate the Indra representation at scale without sacrificing its robustness. From a theoretical view, Kan extensions [13] provide a principled framework for extrapolating from partial distance information to a full representation, preserving maximal structural fidelity.

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
