# OpenReview forum: "The Indra Representation Hypothesis for Multimodal Alignment"
_NeurIPS.cc/2025/Conference — NeurIPS 2025 poster_

### Official Review · Reviewer_7Tyf · 2025-06-06

**Clarity:** 3
**Significance:** 2
**Originality:** 3
**Rating:** 3
**Confidence:** 2

**Summary:**

This paper introduces the "Indra Representation Hypothesis," which states that the representations learned by unimodal foundation models converge towards a shared relational structure. The authors formalize this hypothesis using the V-enriched Yoneda embedding from category theory. This leads to a practical method where a sample's representation is redefined as its vector of distances to all other samples in a given dataset. This distance vector, termed the Indra representation, is computed from the outputs of pre-trained unimodal models. The authors provide theoretical claims that this representation is unique, complete, and structure-preserving. The method is evaluated on cross-model and cross-modal matching tasks involving vision, language, and audio, where it is shown to consistently improve matching accuracy over the original model embeddings.

**Questions:**

1)The proposed method has a computational complexity of $O(n^2d)$ and produces n-dimensional representations. Please comment on the method's viability for large-scale datasets. More importantly, why was the extensive literature on kernel approximation methods, which are designed to mitigate these exact scalability issues, not considered, implemented, or benchmarked against?

2) The theoretical framework guarantees that the Indra representation preserves the structure induced by the cost function d. However, it provides no guidance on how to select d. How does the theory inform this crucial choice? For instance, your ablation shows Euclidean distance also works; does the theory suggest when angular distance might be superior to Euclidean, or vice-versa?

3) Could you refine your claim regarding the "structural myopia" of existing representations? How do you distinguish the relational information captured by your method (inter-sample distances) from the relational information already captured by mechanisms like self-attention (inter-token dependencies)?

**Ethical Concerns:**

["NO or VERY MINOR ethics concerns only"]

**Final Justification:**

The paper presents an interesting hypothesis but does so in a scientific way. It proposes a non-scalable method that is equivalent to computing a kernel matrix without acknowledging or engaging with the literature on kernel methods. It fails to compare against any established baselines, making its empirical claims difficult to assess. The attempt to excuse these omissions by reframing the paper's goals post-submission is not a valid defense. Therefore, I conclude that the paper, in its current form, does not meet the standards for publication.

**Limitations:**

Yes, the authors provided a limitations section in the supplementary material. However, it is wholly inadequate. The authors completely sidestep the critical issue of scalability. Instead, they frame the limitation as their method being "post-training adaptation" and suggest incorporating the ideas into pretraining as future work.  This is not a meaningful limitation but rather a description of the work.

**Quality:**

2

**Strengths And Weaknesses:**

The paper's primary contribution is a novel and elegant conceptualization of representation convergence. The analogy to Indra's Net, combined with the formalization using category theory, provides a thought-provoking new lens through which to view the properties of embeddings from large-scale models.
The proposed method is a simple, post-hoc transformation that requires no retraining of the foundation models. Its generality allows it to be applied to any model that produces vector embeddings, which is a significant practical advantage.
Across the conducted experiments in single-modality and cross-modal settings, the use of the Indra representation demonstrates a modest improvement in matching performance over the baseline of using raw embeddings.

The proposed method is fundamentally non-scalable. To construct representations for a dataset of n samples with embedding dimension d, one must compute an n x n distance matrix, which has a computational complexity of at least $O(n^2d)$ and a memory complexity of $O(n^2)$. The resulting Indra representation for each sample is an n-dimensional vector. This scaling behavior makes the method computationally infeasible for datasets beyond the small-scale validation sets used in the paper (e.g. $n>10^5$). This is, in essence, a kernel method that computes the full kernel matrix. The paper shows a critical lack of rigor by failing to acknowledge this.

The appeal to enriched category theory, while elegant, does not appear to offer concrete guidance for the method's implementation. The Yoneda Lemma guarantees that the mapping to a functor category is fully faithful, preserving all information contained within the chosen cost function d. However, this is a tautology: the representation is complete only with respect to the chosen distance metric. The theory provides no principles for selecting an optimal cost function, which is the single most important design choice in the entire method. Any valid metric space could be used to motivate the same procedure, making the complex theoretical overhead seem more justificatory than generative.
The empirical evaluation is weak because it only compares the Indra representation against the original embeddings. To rigorously assess a new method for post-hoc representation alignment, it must be benchmarked against established techniques. By omitting these comparisons, the paper fails to demonstrate that its proposed method offers a significant advantage over simpler, well-known alternatives.

---

> ### Author Rebuttal · Authors · 2025-07-31
>
> Dear Reviewer 7Tyf,
>
> We sincerely appreciate you for providing thoughtful and constructive comments.
>
> **We are glad to see you recognize our idea as novel and elegant, providing a thought-provoking new lens to view the properties of embeddings of foundation models.**
>
> Below, we address your remaining concerns.
>
> **W1&Q1 Scalability & Complexity**
>
> *[Complexity of Exact Computation]* We agree with you that constructing ***exact*** Indra representations requires a computational complexity of $\mathcal{O}(n^2d)$ and a memory complexity of $\mathcal{O}(n^2)$ for a dataset with $n$ samples and embedding dimension $d$. This quadratic scaling potentially limits the direct applicability of the exact Indra representations to large-scale datasets.
>
> *[Our Research Focus]* However, we respectfully clarify that *the goal of this work is not to propose a new application-driven multi-modal alignment method*. Rather, our focus lies in understanding the fundamental nature of the representations learned by existing unimodal foundation models. Specifically, *we aim to investigate the forms these representations tend to converge to and to identify their ultimate convergent targets*.
>
> In this paper, we argue that unimodal foundation models converge toward a form of Indra representation, which implicitly reflects the relational structure underlying reality (i.e., The Indra Representation Hypothesis). *All our experiments are designed to validate the effectiveness of this form.*
>
> *[Application-Oriented Solutions]* Moreover, ***the scalability concern is addressable in practice***. In the literature, there exists a rich body of work on approximating pairwise distances efficiently. For example, approximate nearest neighbor search (e.g., FAISS, HNSW), landmark-based approximation (e.g., K-means centroids, random subsampling), hashing-based methods (locality sensitive hashing), and sparsified graph constructions (as suggested by Reviewer 6ioa). From an application view, these techniques can be readily adapted to approximate the Indra representation at scale without sacrificing its structural interpretation.
>
> *[Experimental Validation]* To evaluate the effectiveness of approximated Indra representations, we adopt a simple landmark-based approximation via random subsampling. We assess this approach on two tasks: image classification on the CIFAR-100 dataset and image-text matching on the large-scale CC3M dataset. For CIFAR-100, we use the standard data split provided by torchvision.datasets and employ logistic regression (linear probing) to evaluate classification accuracy. For CC3M, we sample 300,000 image-text pairs for evaluation. We extract image features with DINOv2 and text features with RoBERTa. For each image, we compute mean CLIP scores with the top-$k$ candidate texts and compare the performance of the original features against that of the approximated Indra representations.
>
>
> Due to space limitation, ***we kindly refer you to our response to Reviewer yaBd (W1 & Q3) for the experimental details and results.***
>
> **W2&Q2 Distance metric**
>
> We believe there is a misunderstanding regarding the role of enriched category theory in our method. ***The theory is not intended to guide the selection of an optimal cost function for practical applications***. Instead, its purpose is to establish that the Indra representation obtained through our construction is both faithful and complete with respect to the chosen cost function. *The Yoneda Lemma indeed cannot prescribe an optimal cost function for the downstream tasks.*
>
> In practice, *it is not feasible to define a universally optimal cost function across different tasks*. As our experimental results demonstrate, angular distance may outperform Euclidean distance in some settings, while the opposite may be true in others. This highlights that the choice of cost function should be task-dependent, and the theory is not meant to replace empirical validation.
>
> Furthermore, *not all distance functions are valid cost functions in our framework*. To be used as a cost function for the Indra representation, the cost function should satisfy the following properties:
>
> Reflexivity: $d(x, x) = 0$
> Non-negativity: $d(x, y) \in [0, \infty)$
> Triangle inequality: $d(x, z) \leq d(x, y) + d(y, z)$
>
> In summary, *enriched category theory is not used to justify the method post hoc, but to guarantee structural faithfulness and completeness within a mathematically grounded setting.*
>
> **W3& Baselines**
>
> Existing alignment methods, e.g., [50], need additional modules on top of frozen backbones, such as linear projections and adapters, and require substantial fine-tuning on large-scale datasets to learn alignment functions. Once trained, these models become explicitly aligned and are thus more appropriately compared to models like CLIP, which are also designed with explicit alignment.
>
> In contrast, our work aims to investigate the convergent forms and targets of representations learned by unimodal foundation models. Our objective fundamentally differs from that of alignment-driven methods: ***while they actively design and train new modules to enforce alignment, our approach passively explores whether frozen models already contain a more suitable internal representation that supports better cross-modal alignment.***
>
> We uncover that even without any additional training, unimodal foundation models may admit a more optimal representational form than their raw output features. This key difference in motivation, methodology, and training requirements makes a direct empirical comparison with those methods unfair and potentially misleading.
>
> *To the best of our knowledge, this is the first work to demonstrate that frozen representations without any task-specific tuning can achieve improved alignment performance through our proposed representation method, not only within a single modality, but also across vision-language and speech-language modalities.*
>
> **Q1 Kernel approximation methods**
>
> We kindly remind the reviewer that the primary goal of this paper is to investigate the representational forms that unimodal foundation models converge to, and to identify their ultimate convergent targets, rather than to accelerate the proposed method through approximation.
>
> In this work, we hypothesize that these models tend to converge toward a form we define as the Indra representation. Accordingly, all our experiments are designed to validate the effectiveness of this representation, demonstrating that it consistently achieves better alignment than the original model outputs without any additional training.
>
> In fact, *using approximation techniques (e.g., kernel approximation methods) to construct the Indra representation and empirically demonstrate its performance primarily serves as a validation of the Indra representation hypothesis introduced in this paper.*
>
>
> **Q3 Structural myopia & self-attention**
>
> Prior studies on representation convergence often treat the representations produced by foundation models as isolated carriers of information. They analyze convergence purely based on model outputs in a pointwise or instance-level manner. This perspective overlooks the underlying structural relationships embedded within the broader data manifold. In contrast, we argue that model outputs do not directly represent the final converged form, but rather provide the initial scaffold upon which convergence should be understood. We contend that truly convergent representations must be structure-dependent.
>
> While both our method and self-attention involve pairwise relationships, they differ fundamentally in nature and purpose. Our method constructs a representation for each sample based on its static, global relationship to others in the dataset, using a predefined cost function. This results in a complete, non-parametric embedding that faithfully captures the sample's position within the overall data geometry. The representation is invariant across contexts and tasks, and fully determined by the relational structure of the data itself.
>
> In contrast, self-attention models dynamic, context-dependent interactions among tokens, where the strength of each relationship is learned via parameterized query-key-value projections and varies across different inputs and layers. Self-attention embeddings are not faithful to the original data geometry; instead, they adapt to downstream objectives and are shaped by the model's learned parameters.
>
> In short, our method is grounded in global structure and faithful representation, while self-attention captures how a token should change based on its current context and specific tasks.
>
>
> **L1 Limitation section revision**
>
> Thank you for the thoughtful comments. Our experimental results validate the hypothesis by demonstrating improved alignment performance using frozen unimodal foundation models. We believe it is also meaningful to explore this hypothesis in the pretraining stage, to investigate how learning dynamics evolve when the training objectives align with the representational convergent targets.
>
> We have addressed related concerns in the responses above and will revise the limitations section accordingly. In particular, we will clarify our focus on theoretical insights and empirical findings, explicitly acknowledge the computational challenges associated with computing exact Indra representations, and include a discussion of scalable approximation strategies along with the corresponding experimental results.
>
>
> *We hope the above response adequately addresses your concerns. Once again, thank you for your thoughtful feedback and for helping us improve the quality of our paper.*

---

> > ### Comment · Reviewer_7Tyf · 2025-08-04
> > **Response**
> >
> > The authors argue that comparing their method to other alignment techniques (e.g., those using linear projections) is "unfair" because their method is "passive" and requires no training. This argument is unpersuasive.
> >
> > * A simple, learnable linear projection is a standard and essential baseline for any representation alignment task. It establishes a baseline of what is achievable with minimal additional parameters and training.
> > * Other non-parametric, training-free alignment methods exist (e.g., Canonical Correlation Analysis).
> > * By refusing to benchmark against these established techniques, the authors are shielding their method from rigorous evaluation. The claim that their method is superior cannot be substantiated without these comparisons. The burden of proof is on the authors to demonstrate that their computationally expensive $O(n^2)$ method offers benefits over simpler, well-known alternatives. They have not done so.
> >
> > I appreciate the authors' candor in agreeing that the enriched category theory framework does not guide the crucial choice of the distance metric d. However, this confirms my initial criticism: the complex theoretical overhead seems more justificatory than generative. The central guarantee, that the representation is "complete" with respect to the chosen distance metric, is a direct consequence of its construction and can be understood with basic principles of metric geometry. The heavy theoretical machinery does not appear to provide additional insight that justifies its inclusion.

---

> ### Author Response · Authors · 2025-08-07
> **Response Part 1/2**
>
> We thank the reviewer for the thoughtful feedback and **would like to clarify a few points that may have been misunderstood**:
>
> ***1. Our primary goal is not to propose a new alignment method, but rather to demonstrate that the proposed Indra representations exhibit stronger convergence behavior than the original network outputs***.
>
> Specifically, we study the **representation convergence problem**, where we investigate the nature of the representations to which unimodal foundation models tend to converge. Through experiments across various modalities, **we show that the original network outputs may not be the most suitable endpoints for representation convergence. Instead, Indra representations serve as more consistent and convergent targets.**
>
> That said, we agree with the reviewer that including comparisons with standard alignment methods such as learnable linear projections is valuable for contextualizing our findings. In response, **we have now included comparisons with the linear projection baseline [50]**. The classification and matching results are presented in **Tables 1, 2, and 3**, respectively.
>
> As shown in our results, the learnable Linear Projection baseline improves alignment performance compared to the original network outputs, but still underperforms relative to our method. Notably, it achieves stronger results on the image-to-text (I->T) task, because it explicitly projects image features into the text space. However, this forced alignment comes at a cost: aligning image representations to a different modality degrades their utility for classification, particularly under noisy conditions. This indicates reduced robustness and suggests that linear projection may compromise the integrity of the original modality's semantics.
>
> |Table1 Office-Home|A $\sigma$=0.0|A $\sigma$=3.0|A $\sigma$=5.0|A $\sigma$=7.0|C $\sigma$=0.0|C $\sigma$=3.0|C $\sigma$=5.0|C $\sigma$=7.0|P $\sigma$=0.0|P $\sigma$=3.0|P $\sigma$=5.0|P $\sigma$=7.0|R $\sigma$=0.0|R $\sigma$=3.0|R $\sigma$=5.0|R $\sigma$=7.0|
> |--|--|--|--|--|--|--|--|--|--|--|--|--|--|--|--|--|
> |Linear Projection[50]|83.95|48.35|26.34|13.17|84.88|39.52|17.75|08.71|95.38|67.34|38.74|19.93|91.06|72.59|42.89|24.08|
> |ViT|80.25|64.40|44.03|22.63|73.20|50.40|28.64|15.23|92.34|80.74|61.15|35.25|89.22|82.11|60.09|35.32|
> |Ours (Angular)|79.63|65.02|43.62|27.57|69.76|54.98|33.10|18.21|89.75|81.53|64.08|40.77|87.16 |83.49|63.65|40.48|
> |Ours (Euclidean)|80.04 |63.37|43.21|25.51|70.33|55.78|34.48|18.33|89.64 |80.74|64.75|40.77|87.16|83.37|64.56|41.28|
>
> |Table2 CIFAR10|$\sigma$=0.0|$\sigma$=3.0|$\sigma$=5.0|\$\sigma$=7.0|
> |--|--|--|--|--|
> |Linear Projection[50]|85.91|22.29|15.19|12.08|
> |ViT|93.98|87.75|79.77|68.15|
> |Ours (Angular)|94.72|88.08|79.84|68.16|
> |Ours Euclidean|94.84|89.51|80.84|68.71|
>
> |Table3|Top-5|Top-5|Top-10|Top-10|Top-30|Top-30|Top-50|Top-50|
> |--|--|--|--|--|--|--|--|--|
> |MS-COCO|T->I|I->T|T->I|I->T|T->I|I->T|T->I|I->T|
> |Linear Projection[50]|0.460|0.517|0.963|1.052|2.845|3.183|4.890|5.293|
> |ViT+BERT|0.482|0.483|0.967|0.966|2.911|2.905|4.863|4.846|
> |ours|0.663|0.832|1.303|1.613|3.787|4.426|6.199|7.036|
> |NOCAPS|T->I|I->T|T->I|I->T|T->I|I->T|T->I|I->T|
> |Linear Projection[50]|0.496|0.524|1.058|1.042|3.037|3.099|4.938|5.147|
> |ViT+BERT|0.479|0.474|0.956|0.947|2.864|2.844|4.769|4.742|
> |ours|0.701|0.667|1.375|1.293|3.960|3.712|6.449|6.069|
>
> ***2 We would like to clarify that canonical correlation analysis is not a non-parametric, training-free alignment method.***
>
> CCA computes explicit linear projections that map two datasets into a shared latent space.  These projection parameters are derived by solving a generalized eigenvalue problem based on sample covariance matrices, which have a closed-form solution. Although it doesn't involve gradient descent, CCA still involves fitting to the training data and requires estimating projection parameters that depend on the input data.
>
> **We thank the reviewer for suggesting a concrete method for us to compare, and we have added comparisons with CCA**. The results are shown in **Table 4**,  including comparisons with different numbers of canonical components (e.g., CCA-100 refers to using 100 canonical components).
>
> *Please refer to the next page for the continuation of our response.*

---

> ### Author Response · Authors · 2025-08-07
> **Response Part 2/2**
>
> ***3 We respectfully emphasize that our method addresses a fundamentally different research question from the standard alignment approaches referenced by the reviewer.***
>
> Methods such as learnable projections and CCA are explicitly designed to optimize alignment between modalities through training or closed-form solutions. These methods aim to answer the question: **“How can we better align two representations?”**
>
> In contrast, our work investigates the convergent behavior of unimodal foundation models. Specifically, what form their representations tend to converge to. We ask whether the proposed Indra representations can serve as the convergent targets, and naturally yield better alignment without requiring any additional training or projection. That means, our work seeks to answer a distinct question: **“Should Indra representations be considered the final convergent targets of unimodal foundation models?”**
>
>
> |Table4|Top-5|Top-5|Top-10|Top-10|Top-30|Top-30|Top-50|Top-50|
> |--|--|--|--|--|--|--|--|--|
> |MS-COCO|T->I|I->T|T->I|I->T|T->I|I->T|T->I|I->T|
> |ViT+BERT|0.482|0.483|0.967|0.966|2.911|2.905|4.863|4.846|
> |CCA 100|0.498|0.503|0.985|1.001|2.951|3.005|4.934|5.010|
> |CCA 200|0.501|0.498|0.992|1.001|2.974|3.004|4.954|5.001|
> |CCA 300|0.501|0.501|0.991|1.001|2.966|3.006|4.946|5.011|
> |CCA 500|0.487|0.502|0.973|0.999|2.931|2.998|4.859|4.844|
> |Ours|0.663|0.832|1.303|1.613|3.787|4.426|6.199|7.036|
> | | | | | | | | | |
> |ViT+Roberta|0.486|0.491|0.970|0.981|2.912|2.927|4.853|4.874|
> |CCA 100|0.513|0.519|1.014|1.028|2.999|3.019|4.976|5.016|
> |CCA 200|0.510|0.492|1.009|0.992|2.992|2.991|4.982|4.999|
> |CCA 300|0.504|0.489|1.006|0.992|2.992|2.986|4.962|4.981|
> |CCA 500|0.512|0.492|1.021|0.990|3.016|2.975|4.997|4.968|
> |Ours|1.048|0.880|2.065|1.749|5.970|5.149|9.702|8.446|
> | | | | | | | | | |
> |Convnext+BERT|0.396|0.474|0.837|0.950|2.603|2.851|4.412|4.755|
> |CCA 100|0.503|0.495|1.003|0.995|2.986|2.996|4.965|4.994|
> |CCA 200|0.506|0.497|1.006|0.996|2.992|2.984|4.977|4.972|
> |CCA 300|0.509|0.499|1.008|1.001|2.993|2.996|4.985|4.992|
> |CCA 500|0.507|0.496|1.001|0.995|2.991|2.990|4.976|4.980|
> |Ours|0.612|0.537|1.127|1.022|3.182|2.875|5.242|4.783|
> | | | | | | | | | |
> |Convnext+Roberta|0.492|0.480|0.985|0.964|2.962|2.889|4.940|4.824|
> |CCA 100|0.499|0.511|0.993|1.009|2.967|2.986|4.949|4.969|
> |CCA 200|0.481|0.507|0.961|1.007|2.902|2.993|4.869|4.975|
> |CCA 300|0.486|0.509|0.974|1.004|2.910|2.992|4.862|4.982|
> |CCA 500|0.517|0.510|1.025|1.005|3.010|2.999|5.002|4.984|
> |ours|1.005|0.616|1.930|1.217|5.247|3.538|8.267|5.790|
>
>
> ***4 The reviewer’s comment may reflect a misunderstanding of the role that enriched category theory plays in our work.***
>
> Our proposed Indra representation is motivated by the metaphor of Indra’s Net: each sample is defined not in isolation, but through its relational pattern with other samples. The enriched Yoneda embedding formalizes this idea. It shows that **an object in an enriched category is fully and faithfully represented by its morphisms to and from all other objects, with respect to the given enrichment**. In our setting, an object is fully determined by how it relates to all other objects in a category given the cost function. In other words, this theory tells us we can understand an object best by understanding its interactions with all other objects.
>
> We respectfully disagree with the claim that this theoretical structure is merely justificatory. While the completeness result may be partially intuited via classical metric geometry, the enriched Yoneda embedding generalizes beyond metric spaces: **any suitable monoidal structure can serve as the enrichment**. This flexibility is crucial for extending beyond metric embeddings, and the enriched framework ensures that the relational representation is coherent and principled in all such cases.
>
> Indeed, **metric spaces can be viewed as a special case of enriched categories**. While classical metric geometry can ensure that certain representations are complete with respect to a chosen distance function, the enriched Yoneda embedding goes further. It is not merely about comparing objects or measuring distances.  **It’s about reconstructing an object’s identity entirely through its relationships with others, which is central to the philosophy and implementation of our approach**. In contrast, standard metric embeddings only glimpse part of that.
>
>
> *We hope the above response adequately addresses your concerns. Once again, thank you for your thoughtful feedback and for helping us improve the quality of our paper. Please feel free to let us know if any part of our response remains unclear or if you have any further questions!*

---

> > ### Author Response · Authors · 2025-08-08
> > **Follow-Up on Responses to Reviewer's Comments**
> >
> > Dear Reviewer 7Tyf,
> >
> > As the discussion deadline approaches, we would like to kindly ask whether our responses have addressed your remaining concerns.
> >
> > Following your suggestions, we have conducted additional experiments to compare with the learnable linear projection baseline and canonical correlation analysis. The results are provided in Tables 1–4 in Parts 1 and 2 of our above response. We have also further clarified the role of enriched category theory in our work.
> >
> > We really appreciate your thoughtful and constructive comments, and sincerely hope you could check our response. Thank you so much for your time!
> >
> > Sincerely,
> >
> > The Authors

---

### Official Review · Reviewer_6ioa · 2025-06-20

**Clarity:** 3
**Significance:** 2
**Originality:** 2
**Rating:** 4
**Confidence:** 2

**Summary:**

The paper proposes a new theoretical and empirical framework for representation learning inspired by the metaphor of Indra’s Net. The central hypothesis is that unimodal foundation models, despite being trained independently, converge to representations that reflect a shared relational structure of reality. This is formalized using enriched category theory, where each sample is represented by its vector of distances to all other samples under a specific cost function (e.g., angular distance). The resulting Indra representation is theoretically shown to be unique, complete, and structure-preserving. The authors instantiate the method on pretrained models from vision, language, and audio modalities, and evaluate it on cross-model and cross-modal matching tasks. They also test an alternative cost function (Euclidean distance) to assess robustness. The method is applied post hoc, without retraining or supervised alignment.

**Questions:**

1. Have the authors considered evaluating Indra representations on downstream tasks beyond retrieval, such as classification (e.g., via linear probing) or caption generation?
2. Can the authors provide formal or empirical analysis of the time and space complexity of Indra representation computation, especially in large-scale settings?
3. Have the authors considered whether sparse approximations (e.g., k-nearest neighbor graphs) could reduce computational cost while preserving the theoretical guarantees (uniqueness, completeness, structure preservation) of the Indra representation?
4. Can the authors provide qualitative examples or case studies where Indra representations yield more semantically aligned or informative matches than original embeddings?
5. How robust is the Indra representation to the quality of base embeddings? Specifically, how does it behave when the embeddings are noisy or derived from undertrained or small-capacity models?
6. The paper cites prior work on lightweight post-hoc alignment methods, such as linear projections (e.g., [50]), which operate under broadly similar goals. Could the authors elaborate on the decision not to include these as empirical baselines?

**Ethical Concerns:**

["NO or VERY MINOR ethics concerns only"]

**Final Justification:**

The rebuttal addressed most of my concerns, especially regarding the scalability of the method through approximate techniques and the added runtime analysis. The additional experiments on robustness and downstream classification tasks strengthen the empirical evidence for the utility of the Indra representation. While some aspects remain underexplored (ie the qualitative understanding of the captured relations and comparisons to baseline alignment methods), I find the core idea novel and the analysis interesting.

**Limitations:**

Yes, the authors have addressed the limitations and potential negative societal impact of their work. One suggestion for improvement would be to include a discussion of scalability limitations, especially regarding the complexity of computing full pairwise distance matrices.

**Paper Formatting Concerns:**

No major formatting issues observed.

**Quality:**

3

**Strengths And Weaknesses:**

**Strengths**

* The paper introduces a novel and compelling framing of representation learning using the Indra’s Net metaphor.
* The use of category theory and the Yoneda embedding provides an interesting foundation for relational representations.
* The approach is post-hoc and model-agnostic, so it can be applied to any pretrained unimodal encoder without retraining or supervision.
* Empirical validation is provided across multiple modalities (vision, language, audio), model architectures, and datasets, showing consistent performance gains.
* The method is shown to be somewhat robust to the choice of cost function (angular vs. Euclidean distance).

**Weaknesses**
* The method requires computing and storing $n \times n$ distance matrices (Eq. 4), resulting in quadratic complexity in dataset size. The paper does not analyze runtime or memory scaling, nor does it explore sparse or approximate alternatives and their impact on theoretical guarantees.
* The paper could benefit from qualitative visualizations or interpretability analyses to provide intuition on what kinds of relations are being captured.
* The evaluation is limited to retrieval-style tasks using similarity-based matching metrics. The effectiveness of Indra representations on other downstream tasks (e.g., classification, generation, etc) is not explored. Evaluating such tasks would help clarify whether the benefits of the relational structure extend beyond similarity matching.
* The paper cites prior work demonstrating that lightweight post-hoc alignment methods, such as linear projections or Procrustes alignment, can effectively bridge pretrained unimodal encoders (e.g., Merullo et al. [50], Sharma et al. [62]). However, it does not include these as baselines in the empirical evaluation. Comparing against these methods, especially in low-data or zero-shot variants, could clarify whether the gains arise uniquely from the relational structure or could be matched by simpler geometric transformations.

---

> ### Author Rebuttal · Authors · 2025-07-31
>
> Dear Reviewer 6ioa,
>
> **We sincerely appreciate you for providing useful suggestions and for recognizing our idea as novel and compelling**.
>
> Below, we address your remaining concerns.
>
> **W1&Q2&Q3 Complexity**
>
> *[Complexity of Exact Computation]* Due to space limitations, we kindly refer you to our response to Reviewer yaBd (W1 & Q3) for the complexity analysis.
>
> *[Application-Oriented (Scalable) Solutions]* ***The scalability concern is addressable in practice***. In the literature, there exists a rich body of work on approximating pairwise distances efficiently. For example, approximate nearest neighbor search (e.g., FAISS, HNSW), landmark-based approximation (e.g., K-means centroids, random subsampling), hashing-based methods (locality sensitive hashing), and sparsified graph constructions. From an application view, these techniques can be readily adapted to approximate the Indra representation at scale without sacrificing its structural interpretation.
>
> *[Experimental Validation]* Due to space limitations, we kindly refer you to our response to Reviewer yaBd (W1 & Q3) for the corresponding results and analysis.
>
> *[Runtime Analysis]* The table below reports the runtime for constructing Indra representations on the CC3M dataset using a landmark-based approximation with varying numbers of landmarks. All experiments are conducted on an NVIDIA L40S GPU, and each setting is run 20 times to obtain the average runtime.
>
> |#of landmarks|1000|2000|3000|5000|
> |--|--|--|--|--|
> |runtime (s)|0.0476 |0.0666|0.0888|0.1401|
>
> *[Theoretical Guarantees]* Approximate solutions, such as landmark-based methods, may not guarantee uniqueness, completeness, or structure preservation, unless the selected subcategory of landmarks is sufficiently large or dense to distinguish all objects in our sample category. In such cases, we may consider leveraging categorical tools, such as Kan extensions, to extend the behavior of objects defined on the subcategory to the entire category, thereby approximating the full Indra representation. However, this direction lies beyond the scope of the current paper. We will explore this in our future work.
>
>
> **W2 Intuition on captured relations**
>
> Thank you for the helpful suggestion. The types of relations captured by our method are influenced by both the choice of cost function and the foundation model, and are not deterministic. To better understand these relations, we perform visualization analyses using t-SNE and clustering algorithms. Due to rebuttal policy constraints, we are unable to include qualitative visualizations at this stage, but we will incorporate them in the revised version of the paper.
>
> **W3&Q1&Q5 Other downstream tasks & robustness**
>
> We additionally report image classification results on the CIFAR-10, CIFAR-100, and Office-Home datasets under various distance metrics. For CIFAR-10 and CIFAR-100, we use the standard data splits provided by torchvision.datasets. For Office-Home, we evaluate classification accuracy across four domains: Art (A), Clipart (C), Product (P), and Real-World (R), using an 80/20 split for training and testing. Across all datasets, we adopt logistic regression (linear probing) to assess the quality of the extracted representations.
>
> To investigate the robustness of noisy representations, we inject Gaussian noise into the features with varying standard deviations $\sigma  \in \{0.0, 1.5, 3.0, 5.0, 7.0\}$. For each noise level, we perturb the features accordingly and train a linear classifier on the noisy representations. This allows us to assess how classification performance degrades as the feature representations are increasingly corrupted by noise.The corresponding results are presented below.
>
> |cifar10|$\sigma$=0.0|$\sigma$=1.5|$\sigma$=3.0|$\sigma$=5.0|\$\sigma$=7.0| |cifar100|$\sigma$=0.0|$\sigma$=1.5|$\sigma$=3.0|$\sigma$=5.0|\$\sigma$=7.0|
> |--|--|--|--|--|--|--|--|--|--|--|--|--|
> |ViT|93.98|89.60|87.75|79.77|68.15| | |79.45|70.40|54.69|35.76|27.45|
> |Ours (Angular)|**94.72**|**92.90**|**88.08**|**79.84**|**68.16**| | |**80.51**|**74.48**|**61.12**|**42.29**|**28.49**|
> |Ours (Euclidean)|**94.84**|**93.15**|**89.51**|**80.84**|**68.71**| | |**80.09**|**77.36**|**69.00**|**51.59**|**32.74**|
> | | | | | | | | | | | | | |
> |ConvNeXt|97.00|94.37|85.89|80.10|65.85| | |85.77|78.56|62.79|34.39|21.28|
> |Ours (Angular)|**97.20**|**95.22**|**90.77**|**80.18**|**65.93**| | |85.74|**80.30**|**66.04**|**41.58**|**24.45**|
> |Ours (Euclidean)|**97.21**|**96.06**|**92.86**|**81.59**|**66.64**| | |85.64|**82.25**|**72.16**|**51.51**|**30.25**|
> | | | | | | | | | | | | | |
> |DinoV2|99.19|98.58|95.21|85.57|76.54| |  |91.97|89.24|82.21|63.06|40.16|
> |Ours (Angular)|99.19|**98.80**|**96.59**|**87.39**|**76.68**| | |91.77|**89.72**|**83.72**|**67.53**|**47.80**|
> |Ours (Euclidean)|99.14|**98.72**|**96.87**|**89.73**|**77.92**| | |91.93|**89.76**|**84.83**|**74.29**|**58.67**|
>
> |Noise levels|A $\sigma$=0.0|A $\sigma$=3.0|A $\sigma$=5.0|A $\sigma$=7.0|C $\sigma$=0.0|C $\sigma$=3.0|C $\sigma$=5.0|C $\sigma$=7.0|P $\sigma$=0.0|P $\sigma$=3.0|P $\sigma$=5.0|P $\sigma$=7.0|R $\sigma$=0.0|R $\sigma$=3.0|R $\sigma$=5.0|R $\sigma$=7.0|
> |--|--|--|--|--|--|--|--|--|--|--|--|--|--|--|--|--|
> |ViT|80.25|64.40|44.03|22.63|73.20|50.40|28.64|15.23|92.34|80.74|61.15|35.25|89.22|82.11|60.09|35.32|
> |Ours (Angular)|79.63|**65.02**|43.62|**27.57**|69.76|**54.98**|**33.10**|**18.21**|89.75|**81.53**|**64.08**|**40.77**|87.16 |**83.49**|**63.65**|**40.48**|
> |Ours (Euclidean)|80.04 |63.37|43.21|**25.51**|70.33|**55.78**|**34.48**|**18.33**|89.64 |80.74|**64.75**|**40.77**|87.16|**83.37**|**64.56**|**41.28**|
> | | | | | | | | | | | | | | | | | |
> |ConvNeXt|89.71|62.76|27.98|12.14|83.62|54.07|20.85 |09.74|96.62|84.91|44.26|19.37|93.46|82.11|38.30|17.78|
> |ours (Angular)|87.86|59.88|**28.81**|**14.20**|82.70|**57.85**|**25.09**|**11.34**|**96.73**|**85.92**|**45.61**|**22.18**|93.35|**84.63**|**40.71**|**19.61**|
> |Ours (Euclidean)|87.45|58.02|26.75|**12.96**|82.93|**58.30**|**23.83**|**11.23**|96.28|**86.26**|**45.38**|**21.73**|93.35|**84.75**|**40.14**|**19.04**|
> | | | | | | | | | | | | | | | | | |
> |DinoV2|87.65|73.05|46.91|27.78|88.43|75.14|51.09|31.04|96.73|93.24|83.33|60.70|92.78|87.39|71.44|48.51|
> |Ours (Angular)|87.04|70.99|**47.53**|27.37|87.29|**76.63**|**54.75**|**33.56**|96.40|92.79|**84.46**|60.59|**92.89**|**88.53**|**73.17**|**49.89**|
> |ours (Euclidean)|86.21|68.52|44.65|27.16|87.29|**76.75**|**55.21**|**34.59**|96.51|92.57|**84.12**|60.02|92.55|**88.53**|**73.39**|**49.54**|
>
> The results clearly show that stronger backbone models (e.g., DINOv2) lead to better performance for Indra representations across all noise levels. For instance, on CIFAR-100 with $\sigma=0.0$, Indra (Euclidean) achieves 91.93% accuracy using DINOv2 features, compared to 85.64% with ConvNeXt and 80.09% with ViT. This performance gap persists and even widens under higher noise; at σ=7.0\sigma=7.0σ=7.0, Indra (Euclidean) with DINOv2 maintains 58.67%, while ConvNeXt and ViT drop to 30.25% and 32.74%, respectively. In addition, as Gaussian noise increases, our Indra representations consistently retain higher classification accuracy compared to the original features, highlighting their robustness. The performance gains of Indra representations hold across multiple backbone architectures (ViT, ConvNeXt, DINOv2), indicating the broad applicability of the proposed method.
>
> **W4&Q6 Baselines**
>
> The cited methods such as [50] are application-driven with a primary goal of improving cross-modal alignment. They achieve this goal by training additional modules on top of frozen backbones, such as linear projections and adapters. As a result, ***these methods typically require substantial fine-tuning on large-scale datasets to learn optimal alignment mappings***. Once trained, they become explicitly aligned models and are therefore more appropriately compared to models like CLIP, which are also designed with explicit alignment.
>
> In contrast, our work is exploratory in nature, with the main focus on investigating what kind of representation unimodal foundation models inherently converge to. While prior works suggest that different unimodal models exhibit convergence in their output representations, they generally assume that the last-layer outputs themselves constitute this convergent form. We challenge this assumption and hypothesize that there exists a more structured and alignment-friendly representation, i.e., what we propose as the Indra representation. To validate this hypothesis, we compare Indra representations with the original model outputs, showing that Indra representations lead to better alignment performance without any model retraining.
> It is therefore clear that our objective fundamentally differs from that of those methods. While they proactively design and train new modules to enforce alignment, our approach seeks to passively uncover whether existing, frozen models already contain a more suitable representation structure when viewed from a relational perspective.
> In summary, while these alignment methods achieve improved performance through training additional layers, our method reveals that even without training, unimodal foundation models may admit a more optimal representational form than their raw outputs. ***This key difference in motivation, methodology, and training requirements makes a direct empirical comparison with those methods unfair and potentially misleading***.
>
> **Q4 Qualitative examples**
>
> Thank you for the constructive feedback. We have conducted qualitative comparisons demonstrating that Indra representations yield more semantically aligned matches than the original embeddings. Due to rebuttal policy constraints, we are unable to include qualitative examples at this stage, but we will incorporate them in the revised version of the paper.
>
> *We hope the above response adequately addresses your concerns. Once again, thank you for your thoughtful feedback and for helping us improve the quality of our paper.*

---

> ### Author Response · Authors · 2025-08-07
> **Kind Reminder: Approaching Discussion Deadline**
>
> Dear Reviewer 6ioa,
>
> Thank you very much for your time and effort in reviewing our paper.
>
> As the discussion deadline approaches, we would greatly appreciate your thoughts on whether our responses have addressed your concerns. Your feedback is extremely valuable to us, and we would be grateful for any further comments or suggestions you might have.
>
> Thank you once again for your time and consideration!
>
> Sincerely,
>
> The Authors

---

> > ### Comment · Reviewer_6ioa · 2025-08-07
> >
> > I’d like to thank the authors for the detailed rebuttal. The responses are clear, and I appreciate the authors’ efforts in addressing the points and questions I raised. I believe the authors have addressed my concerns. The paper presents an interesting idea, and the revisions have positively influenced my assessment.

---

> > > ### Author Response · Authors · 2025-08-07
> > > **Thank you very much for your positive assessment!**
> > >
> > > Dear Reviewer 6ioa,
> > >
> > > Thank you so much for your positive assessment. We are glad to hear that our response has addressed your concerns!
> > >
> > > We sincerely appreciate your valuable suggestions and constructive feedback, especially your comments on the evaluation of classification tasks and the robustness of the Indra representation. These insights are important in helping us further explore and validate the potential of our proposed approach.
> > >
> > > We truly value your input and will do our best to revise the manuscript accordingly. Once again, thank you for your thoughtful comments and for helping us improve the quality of our paper!
> > >
> > > Best regards,
> > >
> > > The Authors

---

### Official Review · Reviewer_yaBd · 2025-06-23

**Clarity:** 3
**Significance:** 3
**Originality:** 4
**Rating:** 5
**Confidence:** 3

**Summary:**

This paper introduces The Indra Representation Hypothesis, a novel theoretical and empirical framework aimed at understanding representation convergence in unimodal foundation models. Drawing philosophical inspiration from the metaphor of Indra’s Net, the authors propose that foundation models, despite being trained on different modalities and objectives, tend to learn internal representations that converge towards a shared relational structure.

The key contribution is the formalization of this idea through category theory, specifically the V-enriched Yoneda embedding, which leads to the definition of the Indra representation. The Indra representation encodes each sample through its relational profile with respect to all others, using a distance metric (angular distance in this work) between model-generated embeddings. Theoretical guarantees are provided, showing that this representation is unique, complete, and preserves structural relationships.

The paper then instantiates this formulation practically and demonstrates its utility across a wide range of experiments involving cross-model and cross-modal matching tasks, including vision, language, and audio domains. Results show that Indra representations significantly improve matching performance between independently trained unimodal models, outperforming the original embeddings and narrowing the gap with jointly trained multimodal baselines like CLIP and CLAP.

Through both its mathematical grounding and extensive empirical validation, the paper positions the Indra representation as a general, theoretically sound, and practically beneficial approach for understanding and leveraging latent relational structures in foundation models.

**Questions:**

1. Really nice proof! However, the proof of Theory 2, one thing I find confusing, you seem mixing the concept of Identity Property with the Separation Axiom. Would you mind explain what your identity property refers to?
2. Your experiment part is almost excellent. Besides these scores, can you show some real-world downstream task performance of your method?
3. My main concern is that your method has O(N^2) computational complexity. Under this assumption, many of us are interested in how large the computation and storage costs become on truly large-scale datasets. Could you discuss more about that?

**Ethical Concerns:**

["NO or VERY MINOR ethics concerns only"]

**Final Justification:**

The rebuttal experiment has already addressed my concerns. I believe it deserves to be accepted.

**Limitations:**

I suppose you should talk more about limitation.
It fails to address computational cost, storage footprint, sensitivity to the choice of distance metric, and scalability to real downstream tasks, among other issues. Overall, the limitation discussion is too narrow in scope and pays insufficient attention to potential engineering and data challenges.

**Paper Formatting Concerns:**

No.

**Quality:**

3

**Strengths And Weaknesses:**

Strengths:
1. Quality & Clarity: The paper presents a well-grounded and mathematically rigorous framework by introducing the Indra representation using V-enriched category theory. The authors provide theoretical guarantees—uniqueness, completeness, and structure preservation—supported by formal theorems and proofs.

2. Originality: The analogy to Indra’s Net offers a fresh and philosophically motivated perspective on representation learning, distinguishing this work from traditional embedding-centric approaches.

3. Significance: The proposed method is applied to vision, language, and audio models, showing strong performance improvements across cross-model and cross-modal matching tasks, without requiring any retraining or supervision.

Weaknesses:
1. Insufficient limitation discussion: It fails to address computational cost, storage footprint (Indra vectors require pairwise distances over the entire dataset), sensitivity to the choice of distance metric, and scalability to real downstream tasks, among other issues. Overall, the limitation discussion is too narrow in scope and pays insufficient attention to potential engineering and data challenges.

2. Small Unclear in proof of theory 2: Seems missing proof of separation axiom (d(x,y)=0 ⇒ x=y)

---

> ### Author Rebuttal · Authors · 2025-07-31
>
> Dear Reviewer yaBd,
>
> **We sincerely appreciate you for providing helpful suggestions and for recognizing our idea as fresh and distinguishable from traditional approaches**.
>
> Below, we address your remaining concerns.
>
> **W1&Q3 Limitation discussion**
>
> *[Complexity of Exact Computation]* Constructing ***exact*** Indra representations requires a computational complexity of $\mathcal{O}(n^2d)$ and a memory complexity of $\mathcal{O}(n^2)$ for a dataset with $n$ samples and embedding dimension $d$. This quadratic scaling potentially limits the direct applicability of the exact Indra representations to large-scale datasets.
>
> *[Application-Oriented (Scalable) Solutions]* ***The scalability concern is addressable in practice***. In the literature, there exists a rich body of work on approximating pairwise distances efficiently. For example, approximate nearest neighbor search (e.g., FAISS, HNSW), landmark-based approximation (e.g., K-means centroids, random subsampling), hashing-based methods (locality sensitive hashing), and sparsified graph constructions (as suggested by Reviewer 6ioa). From an application view, these techniques can be readily adapted to approximate the Indra representation at scale without sacrificing its structural interpretation.
>
> *[Experimental Validation]* To evaluate the effectiveness of ***approximated*** Indra representations, we adopt a simple landmark-based approximation via random subsampling. We randomly select $m < n$ landmarks as representative samples, reducing the computational complexity to $\mathcal{O}(nmd)$ and the memory complexity to $\mathcal{O}(nm)$. We assess this approach on two tasks: image classification on the CIFAR-100 dataset and image-text matching on the large-scale CC3M dataset.
>
> For CIFAR-100, we use the standard data split provided by torchvision.datasets and employ logistic regression (linear probing) to evaluate classification accuracy. To assess the robustness, we inject Gaussian noise with different levels of standard deviation $\sigma$ into the extracted features.
>
> For CC3M, we sample 300,000 image-text pairs for evaluation. We extract image features with DINOv2 and text features with RoBERTa. For each image, we compute mean CLIP scores with the top-$k$ candidate texts and compare the performance of the original features against that of the approximated Indra representations.
>
> Results under varying numbers of landmarks are presented below.
>
> |CIFAR-100|$\sigma$=0.0|$\sigma$=1.5|$\sigma$=3.0|$\sigma$=5.0|\$\sigma$=7.0||CIFAR-100|$\sigma$=0.0|$\sigma$=1.5|$\sigma$=3.0|$\sigma$=5.0|\$\sigma$=7.0|# of landmarks|
> |--|--|--|--|--|--|--|--|--|--|--|--|--|:--:|
> |DinoV2|91.97|89.24|82.21|63.06|40.16||DinoV2|91.97|89.24|82.21|63.06|40.16|---|
> |Ours (Euclidean)|90.46|88.56|82.49|66.00|40.46||Ours (Angular) |91.75|89.55|84.03|67.83|46.90|1000|
> |Ours (Euclidean)|91.83|90.08|84.31|71.38|49.33||Ours (Angular) |92.09|90.25|85.37|72.02|53.66|2000|
> |Ours (Euclidean)|91.85|89.28|84.91|70.39|52.02||Ours (Angular) |**92.34**|**90.34**|**85.53**|**73.74**|**55.18**|3000|
> |Ours (Euclidean)|92.06|**90.12**|85.81|71.98|54.56||Ours (Angular) |92.10|90.20|85.48|73.40|55.15|5000|
> |Ours (Euclidean)|**92.20**|**90.12**|**85.56**|73.23|56.91||Ours (Angular) |92.24|89.99|85.14|72.03|53.68|10000|
>
>
> ||#of landmarks|k=5|k=10|
> |--|--|--|--|
> |DinoV2+RoBERTa|---|25219.78|25208.54|
> |Ours |5000|28225.07|26836.45|
>
> The experimental results demonstrate that ***the approximated Indra representations still achieve better classification and alignment performance for real-world applications***. Specifically, the Indra representations consistently outperform the original representations under increasing levels of Gaussian noise, particularly at higher noise levels, highlighting their superior robustness. Furthermore, increasing the number of landmarks may make improvements in classification accuracy.
>
> **W2 &Q1 Proof of Theory 2**
>
> Thank you for carefully reviewing the proof. You are absolutely right, our statement that “the cost function $d$ satisfies the identity property” should be revised to “the cost function $d$ satisfies the T₀ separation axiom.” The identity property, which requires that $d(X_i,X_j)=0$ does not imply $X_i=X_j$. We will revise the theorem and proof accordingly to incorporate a stricter condition on the cost function.
>
> **Q2 Downstream tasks**
>
> We additionally report image classification results on the CIFAR-10, CIFAR-100, and Office-Home datasets under various distance metrics. For CIFAR-10 and CIFAR-100, we use the standard data splits provided by torchvision.datasets. For Office-Home, we evaluate classification accuracy across four domains: Art (A), Clipart (C), Product (P), and Real-World (R), using an 80/20 split for training and testing. Across all datasets, we adopt logistic regression (linear probing) to assess the quality of the extracted representations. To evaluate robustness, we inject Gaussian noise into the features with varying standard deviations $\sigma$. The corresponding results are presented below.
>
>
> |cifar10|$\sigma$=0.0|$\sigma$=1.5|$\sigma$=3.0|$\sigma$=5.0|\$\sigma$=7.0| |cifar100|$\sigma$=0.0|$\sigma$=1.5|$\sigma$=3.0|$\sigma$=5.0|\$\sigma$=7.0|
> |--|--|--|--|--|--|--|--|--|--|--|--|--|
> |ViT|93.98|89.60|87.75|79.77|68.15| | |79.45|70.40|54.69|35.76|27.45|
> |Ours (Angular)|**94.72**|**92.90**|**88.08**|**79.84**|**68.16**| | |**80.51**|**74.48**|**61.12**|**42.29**|**28.49**|
> |Ours (Euclidean)|**94.84**|**93.15**|**89.51**|**80.84**|**68.71**| | |**80.09**|**77.36**|**69.00**|**51.59**|**32.74**|
> | | | | | | | | | | | | | |
> |ConvNeXt|97.00|94.37|85.89|80.10|65.85| | |85.77|78.56|62.79|34.39|21.28|
> |Ours (Angular)|**97.20**|**95.22**|**90.77**|**80.18**|**65.93**| | |85.74|**80.30**|**66.04**|**41.58**|**24.45**|
> |Ours (Euclidean)|**97.21**|**96.06**|**92.86**|**81.59**|**66.64**| | |85.64|**82.25**|**72.16**|**51.51**|**30.25**|
> | | | | | | | | | | | | | |
> |DinoV2|99.19|98.58|95.21|85.57|76.54| |  |91.97|89.24|82.21|63.06|40.16|
> |Ours (Angular)|99.19|**98.80**|**96.59**|**87.39**|**76.68**| | |91.77|**89.72**|**83.72**|**67.53**|**47.80**|
> |Ours (Euclidean)|99.14|**98.72**|**96.87**|**89.73**|**77.92**| | |91.93|**89.76**|**84.83**|**74.29**|**58.67**|
>
> |Noise levels|A $\sigma$=0.0|A $\sigma$=3.0|A $\sigma$=5.0|A $\sigma$=7.0|C $\sigma$=0.0|C $\sigma$=3.0|C $\sigma$=5.0|C $\sigma$=7.0|P $\sigma$=0.0|P $\sigma$=3.0|P $\sigma$=5.0|P $\sigma$=7.0|R $\sigma$=0.0|R $\sigma$=3.0|R $\sigma$=5.0|R $\sigma$=7.0|
> |--|--|--|--|--|--|--|--|--|--|--|--|--|--|--|--|--|
> |ViT|80.25|64.40|44.03|22.63|73.20|50.40|28.64|15.23|92.34|80.74|61.15|35.25|89.22|82.11|60.09|35.32|
> |Ours (Angular)|79.63|**65.02**|43.62|**27.57**|69.76|**54.98**|**33.10**|**18.21**|89.75|**81.53**|**64.08**|**40.77**|87.16 |**83.49**|**63.65**|**40.48**|
> |Ours (Euclidean)|80.04 |63.37|43.21|**25.51**|70.33|**55.78**|**34.48**|**18.33**|89.64 |80.74|**64.75**|**40.77**|87.16|**83.37**|**64.56**|**41.28**|
> | | | | | | | | | | | | | | | | | |
> |ConvNeXt|89.71|62.76|27.98|12.14|83.62|54.07|20.85 |09.74|96.62|84.91|44.26|19.37|93.46|82.11|38.30|17.78|
> |ours (Angular)|87.86|59.88|**28.81**|**14.20**|82.70|**57.85**|**25.09**|**11.34**|**96.73**|**85.92**|**45.61**|**22.18**|93.35|**84.63**|**40.71**|**19.61**|
> |Ours (Euclidean)|87.45|58.02|26.75|**12.96**|82.93|**58.30**|**23.83**|**11.23**|96.28|**86.26**|**45.38**|**21.73**|93.35|**84.75**|**40.14**|**19.04**|
> | | | | | | | | | | | | | | | | | |
> |DinoV2|87.65|73.05|46.91|27.78|88.43|75.14|51.09|31.04|96.73|93.24|83.33|60.70|92.78|87.39|71.44|48.51|
> |Ours (Angular)|87.04|70.99|**47.53**|27.37|87.29|**76.63**|**54.75**|**33.56**|96.40|92.79|**84.46**|60.59|**92.89**|**88.53**|**73.17**|**49.89**|
> |ours (Euclidean)|86.21|68.52|44.65|27.16|87.29|**76.75**|**55.21**|**34.59**|96.51|92.57|**84.12**|60.02|92.55|**88.53**|**73.39**|**49.54**|
>
>
>
> **L1 Limitation section revision**
>
> Thank you for the constructive comments. We have addressed these potential issues in the discussion above and will revise the limitations section accordingly. Specifically, we will explicitly acknowledge the computational challenges of computing exact Indra representations and include a discussion of scalable approximation strategies, supported by corresponding experimental validation.
>
>
> *We hope the above response adequately addresses your concerns. Once again, thank you for your thoughtful feedback and for helping us improve the quality of our paper.*

---

> > ### Comment · Reviewer_yaBd · 2025-08-05
> >
> > I thank the authors for their comprehensive response. I think the response addresses my concerns. This is an interesting idea, and I believe it deserves to be accepted, so I decided to raise the score to 5.

---

> ### Author Response · Authors · 2025-08-06
> **Thank you very much for recommending acceptance!**
>
> Dear Reviewer yaBd,
>
> Thank you very much for recommending acceptance!
>
> We truly appreciate your recognition of our novelty and your thoughtful feedback. Your comments are highly valuable, and we will do our best to revise the manuscript accordingly. Once again, thank you for helping us improve the quality of our work!
>
> Best regards,
>
> The Authors

---

### Official Review · Reviewer_enB4 · 2025-07-03

**Clarity:** 4
**Significance:** 2
**Originality:** 4
**Rating:** 3
**Confidence:** 3

**Summary:**

Motivated by recent findings that unimodal foundation models often exhibit strong correlations between their feature representations, this paper introduces the concept of the Indra Representation. Inspired by the ancient notion of Indra’s Net—a metaphysical idea from Eastern philosophy suggesting that each entity is a reflection of all others—the authors formalize a representation that captures the relational structure underlying model features. This formulation provides a unified framework for interpreting and aligning features across different models and modalities. The effectiveness of the proposed representation is validated through experiments conducted across diverse datasets and settings.

**Questions:**

You can directly refer to the weaknesses part.

**Ethical Concerns:**

["NO or VERY MINOR ethics concerns only"]

**Limitations:**

yes

**Quality:**

3

**Strengths And Weaknesses:**

Strengths:
1.	Building upon the ancient philosophical concept of Indra’s Net as a foundational metaphor, this work formalizes and instantiates it within a modern machine learning framework. I find this conceptual grounding to be highly novel and intellectually compelling.
2.	The paper presents comprehensive experiments demonstrating that the proposed Indra representation achieves stronger alignment across features from different unimodal foundation encoders, compared to using raw features directly from each encoder. This provides solid empirical support for the effectiveness of the proposed framework.

Weaknesses:
1.	While the proposed representation is theoretically elegant and empirically validated, the paper could benefit from a clearer articulation of its practical utility. Specifically, it remains somewhat unclear in what real-world scenarios such cross-model, cross-modality feature alignment is necessary. Furthermore, the authors do not address why existing well-aligned models, such as CLIP or more recent MLLM like GPT-4o, would not already suffice in such contexts. Providing stronger motivation for why this new representation is needed in practice would strengthen the impact of the work.
2.	The use of number notes such as ① and ② throughout the paper feels informal and somewhat inconsistent with the typical academic writing style expected in formal publications. It may be advisable to adopt more standard notation or formatting for clarity and professionalism.
3.	While the proposed method shows strong performance on the selected benchmarks, the experiments are primarily conducted on datasets with a limited number of categories. In real-world applications, it may be challenging to define the appropriate scope and number of relevant entities for constructing the Indra representation, potentially limiting its scalability and practicality in large-scale or open-domain settings.

---

> ### Author Rebuttal · Authors · 2025-07-31
>
> Dear Reviewer enB4,
>
>
> **We sincerely appreciate you for providing constructive comments and are glad to see you recognize our idea as highly novel and intellectually compelling.**
>
> Below, we address your remaining concerns.
>
> **W1 Practical utility**
>
> Thank you for providing constructive suggestions. We would like to clarify that the primary goal of this work is not to demonstrate that our proposed Indra representations outperform well-aligned models such as CLIP or GPT-4o in practical applications, nor is it to argue that such models are insufficient.
>
> Instead, our core contribution is to investigate what form the learned representations of large-scale unimodal foundation models converge to, after training on unimodal data. Prior studies have shown that such models tend to converge in representation space, but they do not explain what the representations ultimately converge to. Our work offers a new representation convergence hypothesis and provides empirical results to validate this hypothesis.
>
> The ***practical utility*** of our proposed Indra representation lies in its ability to improve cross-modality alignment without the need of additional adapter or large-scale finetuning. In domains like medicine or other specialized fields, there may not exist multimodal foundation models, nor is it always feasible to train them. In such settings, leveraging existing unimodal models through our Indra representation offers a compelling solution for cross-modal tasks.
>
> As demonstrated in our experiments, the proposed Indra representation improves performance not only in unimodal tasks, but also across vision-language and speech-language modalities. Furthermore, the added classification experiments show that our representation also enhances classification accuracy and robustness (kindly refer to our response to Reviewer yaBd (W1 & Q3)).
>
> In summary, *our work is not intended to replace well-aligned models but to uncover a foundational mechanism that explains how unimodal models can be structurally leveraged or adapted for cross-modal settings, especially where aligned models are unavailable or infeasible.*
>
>
> **W2 Notation**
>
> We appreciate the reviewer’s feedback regarding the use of numbered notes. We understand that this stylistic choice may come across as informal or inconsistent with conventional academic norms. We have revised the manuscript to adopt a more standard academic style. We thank the reviewer for highlighting this and will ensure the final version adheres to expected standards.
>
>
> **W3 Scalability and practicality**
>
> *[Application-Oriented (Scalable) Solutions]* ***The scalability concern is addressable in practice***. In the literature, there exists a rich body of work on approximating pairwise distances efficiently. For example, approximate nearest neighbor search (e.g., FAISS, HNSW), landmark-based approximation (e.g., K-means centroids, random subsampling), hashing-based methods (locality sensitive hashing), and sparsified graph constructions (as suggested by Reviewer 6ioa). From an application view, these techniques can be readily adapted to approximate the Indra representation at scale without sacrificing its structural interpretation.
>
> *[Experimental Validation]* To evaluate the effectiveness of approximated Indra representations, we adopt a simple landmark-based approximation via random subsampling. We randomly select $m < n$ landmarks as representative samples, reducing the computational complexity to $\mathcal{O}(nmd)$ and the memory complexity to $\mathcal{O}(nm)$. We assess this approach on two tasks: image classification on the CIFAR-100 dataset and image-text matching on the large-scale CC3M dataset.
>
> For CIFAR-100, we use the standard data split provided by torchvision.datasets and employ logistic regression (linear probing) to evaluate classification accuracy. To assess the robustness, we inject Gaussian noise with different levels of standard deviation $\sigma$ into the extracted features.
>
> For CC3M, we sample 300,000 image-text pairs for evaluation. We extract image features with DINOv2 and text features with RoBERTa. For each image, we compute mean CLIP scores with the top-$k$ candidate texts and compare the performance of the original features against that of the approximated Indra representations.
>
> Results under varying numbers of landmarks are presented below.
>
> |CIFAR-100|$\sigma$=0.0|$\sigma$=1.5|$\sigma$=3.0|$\sigma$=5.0|\$\sigma$=7.0||CIFAR-100|$\sigma$=0.0|$\sigma$=1.5|$\sigma$=3.0|$\sigma$=5.0|\$\sigma$=7.0|# of landmarks|
> |--|--|--|--|--|--|--|--|--|--|--|--|--|:--:|
> |DinoV2|91.97|89.24|82.21|63.06|40.16||DinoV2|91.97|89.24|82.21|63.06|40.16|---|
> |Ours (Euclidean)|90.46|88.56|82.49|66.00|40.46||Ours (Angular) |91.75|89.55|84.03|67.83|46.90|1000|
> |Ours (Euclidean)|91.83|90.08|84.31|71.38|49.33||Ours (Angular) |92.09|90.25|85.37|72.02|53.66|2000|
> |Ours (Euclidean)|91.85|89.28|84.91|70.39|52.02||Ours (Angular) |**92.34**|**90.34**|**85.53**|**73.74**|**55.18**|3000|
> |Ours (Euclidean)|92.06|**90.12**|85.81|71.98|54.56||Ours (Angular) |92.10|90.20|85.48|73.40|55.15|5000|
> |Ours (Euclidean)|**92.20**|**90.12**|**85.56**|73.23|56.91||Ours (Angular) |92.24|89.99|85.14|72.03|53.68|10000|
>
>
> ||#of landmarks|k=5|k=10|
> |--|--|--|--|
> |DinoV2+RoBERTa|---|25219.78|25208.54|
> |Ours |5000|28225.07|26836.45|
>
> The experimental results demonstrate that ***the approximated Indra representations still achieve better classification and alignment performance for real-world applications***. Specifically, the Indra representations consistently outperform the original representations under increasing levels of Gaussian noise, particularly at higher noise levels, highlighting their superior robustness. Furthermore, increasing the number of landmarks may make improvements in classification accuracy.
>
>
> *We hope the above response adequately addresses your concerns. Once again, thank you for your thoughtful feedback and for helping us improve the quality of our paper!*

---

> ### Author Response · Authors · 2025-08-07
> **Kind Reminder: Approaching Discussion Deadline**
>
> Dear Reviewer enB4,
>
> Thank you very much for your time and effort in reviewing our paper.
>
> As the discussion deadline approaches, we would greatly appreciate your thoughts on whether our responses have addressed your concerns and if they have helped in re-evaluating our work. Your feedback is extremely valuable to us, and we would be grateful for any further comments or suggestions you might have.
>
> Thank you once again for your time and consideration!
>
> Sincerely,
>
> The Authors

---

### Author Response · Authors · 2025-08-03
**Kind Reminder: Discussion Deadline Approaching**

**Dear Reviewers**,

Thank you very much for your thoughtful and constructive feedback on our submission.

We are encouraged to see a consistent recognition of this paper’s novelty across all reviews:

- **Reviewer enB4**: *"Highly novel and intellectually compelling."*

- **Reviewer yaBd**: *"Fresh and philosophically motivated, distinguishable from traditional approaches."*

- **Reviewer 6ioa**: *"Novel and compelling, an interesting foundation for relational representations."*

- **Reviewer 7Tyf**: *"A novel and elegant conceptualization, providing a thought-provoking new lens."*





To facilitate your re-engagement with the paper and address any remaining concerns, we briefly summarize the **core contributions** and **new experimental findings** below:

---

**Core Contributions**

1. **The Indra Representation Hypothesis:**
Drawing inspiration from the philosophical metaphor of Indra’s Net, we propose that unimodal foundation models naturally converge toward a form of representation that implicitly reflects the underlying relational structure of reality.


2. **Formalization via Enriched Category Theory:**
We formalize these representations using V-enriched Yoneda embeddings, ensuring that the resulting representations are unique, complete, and structure-preserving.


3. **Empirical Validation Across Modalities:**
Extensive experiments demonstrate that Indra representations lead to improved alignment performance without retraining. This effect generalizes beyond unimodal models to vision-language and speech-language settings.

---

**New Findings in Response to Reviewer Suggestions**

In response to reviewer feedback, we conducted additional experiments on new tasks and settings. Notably, we find that:

- Indra representations improve classification performance, demonstrating **increased accuracy and robustness even under highly noisy conditions**.

- Landmark-based approximations remain effective on large-scale datasets, **enhancing the practicality and scalability** of the proposed Indra representations.

---

We hope these clarifications and additional results address your concerns and further support the significance of our contributions.
***As the discussion period approaches, we would be very grateful if you could take a moment to review our responses.***

We sincerely appreciate your time and dedication in helping improve the quality of our work!

Warm regards,

The Authors

---

### Decision · Program_Chairs · 2025-09-17

**Decision:**

Accept (poster)

**Comment:**

The paper proposes the Indra Representation Hypothesis: unimodal models converge to a relational structure formalized with a V-enriched Yoneda embedding. It builds distance-based “Indra” vectors and shows post-hoc gains for cross-model and cross-modal matching, plus robustness, using landmark/ANN approximations. Two reviewers praise the novelty and the stronger rebuttal, and recommend accept or borderline accept. Two others question practical need versus multimodal models, O(n²) cost and storage, and guidance on metric choice; one barely engaged.

I recommend accept, but this is borderline from my perspective. The idea is fresh and now better supported by added baselines (linear projection, CCA) and runtime/large-scale results. Scaling and metric-choice risks remain, but approximations and sensitivity checks help, and the camera-ready should expand limitations and fix notation. With split reviews and moderate confidence, I still see clear value for the community.